# POLICY OPTIMIZATION WITH EXPERIENCE REPLAY: GUIDING REASONING MODELS TO COMPLETE THE REASONING PATH

## ABSTRACT

To our knowledge, in the field of large language models, all existing reinforcement fine-tuning algorithms require generating a complete reasoning process starting from the question, which results in a substantial time overhead during the rollout phase of training.Challenging this conventional approach, we propose the assumption that during reinforcement fine-tuning, the model only needs to generate part of the reasoning process. We analyze the impact of different segments of the reasoning path on the correctness of the final result, and based on these insights, we introduce **Policy Optimization with Experience Replay (POER)**, a plug-and-play reinforcement fine-tuning algorithm. Unlike traditional reinforcement fine-tuning algorithms that generate full reasoning paths, POER trains the model by generating suffixes of the reasoning path using experience caching, thereby significantly reducing training time while improving training stability.From evaluations during the rollout phase of training, POER reduces token generation in this phase by approximately 95%, greatly lowering the theoretical time overhead. In practical training, compared with full-path reinforcement fine-tuning algorithms, POER reduces the training time of the 1.5B model by 90% and the 7B model by 72%, while maintaining performance comparable to typical algorithms such as GRPO and DAPO. We have open-sourced the code in an anonymous repository: `https://anonymous.4open.science/r/POER-4BF2`

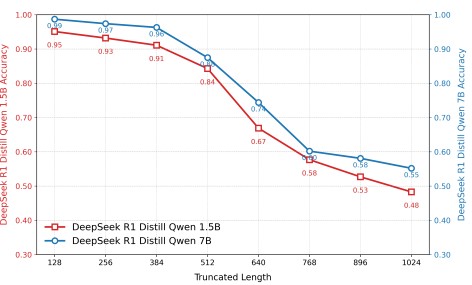 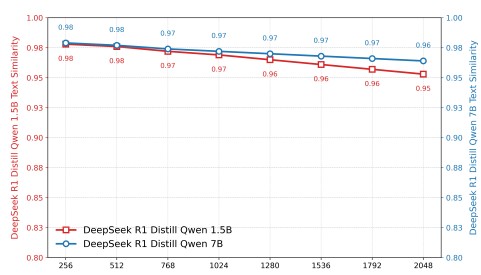

Figure 1: The left figure shows the **DeepSeekR1-Qwen-Distill-7b** and **DeepSeekR1-Qwen-Distill-1.5b** models. For each question, an initial answer is generated and then truncated; from the truncation point, 256 answers are subsequently generated, and the relationship between truncation length and the overall average accuracy is analyzed. The right figure shows 256 answers generated for each training question. Answers exceeding 2048 tokens are selected, and **BERT** is used to measure the similarity between equal-length prefix segments. The similarity metric is defined as:$\text{sim} = \frac{2}{n(n-1)} \sum_{i=1}^{n-1} \sum_{j=i+1}^{n} \frac{\text{BERT}(s_i) \cdot \text{BERT}(s_j)^{\top}}{\|\text{BERT}(s_i)\| \, \|\text{BERT}(s_j)\|}$

## 1 INTRODUCTION

In recent years, large language models (LLMs) (OpenAI et al., 2024b; Touvron et al., 2023; Zeng et al., 2023) have achieved remarkable breakthroughs in reasoning and generalization capabilities (Wang et al., 2025), particularly after the introduction of reinforcement learning (RL) during the post-training

stage (Ouyang et al., 2022). Pioneering works such as OpenAI's O1 (OpenAI et al., 2024a) and DeepSeek-R1 (DeepSeek-AI et al., 2025) have demonstrated impressive reasoning-time efficiency, primarily due to the synergistic combination of reinforcement learning and chain-of-thought (CoT) reasoning (Wei et al., 2023). This paradigm shift highlights the transformative potential of RL-based post-training in pushing the boundaries of LLM performance.

Despite its promising prospects, applying reinforcement learning in post-training remains immature and highly challenging, with numerous obstacles hindering its widespread adoption. Regarding time overhead, RL fine-tuning typically generates many samples during the sampling stage. However, parameter updates cannot proceed until all samples are completed, leading to significant underutilization of computational resources. Furthermore, during RL fine-tuning of language models, rewards are computed only after generating the final token based on task-specific criteria. This paradigm, known as Reinforcement Learning with Verifiable Rewards (RLVR) (Lambert et al., 2025), lacks intermediate feedback and produces sparse rewards. Such sparsity hinders the model's ability to learn optimal policies and contributes to training instability (Lightman et al., 2023).

Most current research efforts addressing these challenges focus on optimizing the policy gradient function (Sutton et al., 1999), which has achieved some success. However, these approaches often overlook the critical role of sampling strategies.

We observe that a significant underlying issue stems from the policy model's need to identify a reasoning trajectory from the beginning of the problem to the correct answer. This approach—comparing entire reasoning paths using policy gradients—leads to excessive randomness during the sampling phase. Although it expands the search space, it often fails to find suitable reasoning paths, resulting in inefficient sampling and high variance.

An alternative perspective arises: since exploring a complete reasoning path from the beginning of the problem introduces various drawbacks, why not train the policy model to complete a reasoning path based on **partially correct reasoning process hint** instead? We found that this is feasible. Through our experiments, enabling the model to complete correct reasoning paths can still effectively teach it to generate whole reasoning trajectories from the initial problem statement. Based on this insight, we propose the *Policy Optimization with Experience Replay*(POER). Our method is grounded in a reasonable assumption: the early tokens of a reasoning path that leads to the correct answer are more likely to guide the model toward the correct reasoning trajectory. Furthermore, we investigate the relationship between the length of the truncated trailing tokens and the model's generation accuracy. The results confirm that the initial tokens of correct answers play a crucial role in steering the model toward correct solutions, and that longer prefix lengths positively correlate with higher generation accuracy.

Specifically, we construct a cache pool for the GRPO to store previously generated reasoning paths and continuously update it during training. After we complete the sampling generation stage for each question, we add the reasoning path that leads to the correct answer into the cache. When we later reencounter the same question, we retrieve the first **n** tokens of the corresponding reasoning path from the cache, prepend them to the prompt, and then perform sampling. Experimental results show that this method is plug-and-play, improves training stability during the RL stage, significantly reduces the policy model's sampling time cost, and achieves notable performance gains.

**Contributions** We propose POER, a novel framework for reinforcement fine-tuning of LLMs, introducing an experience replay mechanism in the sampling stage. Key advantages are: **plug-and-play**: easily integrates into other RL fine-tuning methods; **reduced resource consumption**: up to 92.6% faster training; **strong stability**: mitigates common RL instability in reasoning models.

We evaluate POER on Deepseek-R1-Distill-Qwen 1.5B and 7B across six datasets. Results show around 90% reduction in training time, a 2% performance improvement over GRPO and DAPO, and support for mini-batch, multi-step updates.

## 2 RELATED WORKS

**Reinforcement Fine-Tuning** Reinforcement Fine-Tuning (RFT) guides the model fine-tuning process through the reward mechanisms of reinforcement learning, greatly enhancing generalization and accuracy. Kimi v1.5 (Team et al., 2025) and ReFT (Luong et al., 2024) employ traditional

Proximal Policy Optimization (PPO) (Schulman et al., 2017) for RFT and have demonstrated excellent performance. DeepSeek-R1 (DeepSeek-AI et al., 2025) adopts GRPO and uses verifiable reward strategies to compute policy gradients directly. DAPO (Yu et al., 2025b) further optimizes GRPO to improve training stability. R1-V (Chen et al.), VLM-R1 (Shen et al., 2025a), and LMM-R1 (Peng et al., 2025) extend RFT into the multi-modal domain. While many reinforcement learning algorithms suffer from reward saturation as training steps increase, leading to reward hacking (Eisenstein et al., 2024). Satori (Shen et al., 2025b) uses SFT distillation to mitigate it, and O1-Prune (Luo et al., 2025) employs post-hoc length pruning to enhance stability.

**Efficient Inference** Test-time scaling (Chen et al., 2024) significantly increases training and inference costs, as models tend to generate lengthy reasoning chains. To reduce the time cost of RL training, UPFT (Ji et al., 2025) proposes fine-tuning the model using only the first n tokens. However, it is impossible to validate reasoning accuracy properly. ThinkPrune (Hou et al., 2025) sets a length constraint during RL training to limit the model's thinking length and reduce inference costs. O1-Prune (Luo et al., 2025) enhances training stability through length pruning and reduces the high cost associated with long reasoning chains. Hao et al. (2024) optimizes inference by compressing lengthy reasoning chains into latent space, while Chen et al. (2025); Yu et al. (2025a) reduces inference costs by aggregating tokens.

## 3 METHOD

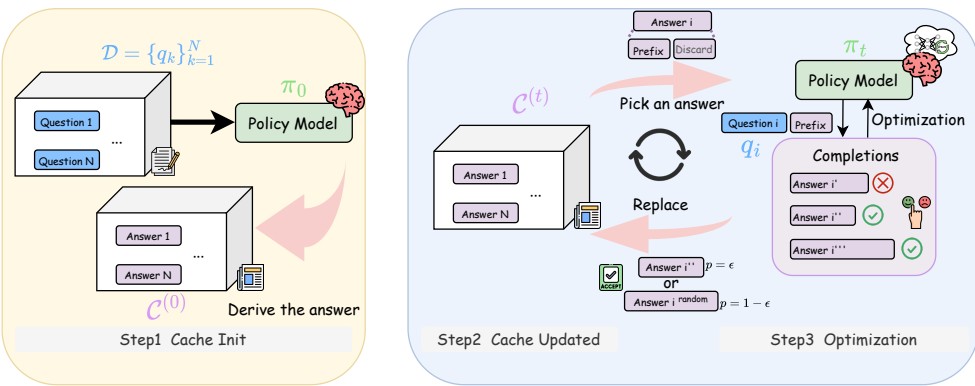

Figure 2: Overview of the POER framework. The entire training process is described as follows: Cached answer fragments are used by the model to generate new responses; either the best or a random response is selected based on the reward system for optimization; and the cache is continuously updated to improve training efficiency and stability.

### 3.1 CACHE POOL INITIALIZATION

First, we denote the dataset of samples as $\mathcal{D} = \{q_k\}_{k=1}^{N}$, where $q_k$ represents the $k$-th question in the dataset. We denote the initial model parameters as $\theta_0$, and we represent the model's answering policy by $\pi_{\theta_0}$. Before training begins, we initialize the cache pool as $\mathcal{C}^{(0)}$ as follows:

$$\mathcal{C}^{(0)} = \{(q_k, a_k) \mid a_k \sim \pi_{\theta_0}(\cdot|q_k), \forall q_k \in \mathcal{D}\} \tag{1}$$

This stage uses the initial model policy to sample the dataset $\mathcal{D}$. To retrieve the response $a_k$ corresponding to question $q_k$ from the cache pool, we define the retrieval operation as:

$$a_k := \{a \mid (q_k, a) \in \mathcal{C}\}. \tag{2}$$

Here, $a_k$ denotes the answer associated with question $q_k$ in the cache pool $\mathcal{C}$.

## 3.2 TRUNCATED ANSWER SAMPLING OPTIMIZATION

**Sampling Generation**    At each sampling stage, we use the POER strategy to retrieve the historical response $a_k$ for each question $q_k$ from the cache pool $\mathcal{C}$. We then remove the last $m$ tokens and concatenate the remaining prefix with $q_k$ to generate a new response $o$. We express this process as:

$$o = a_k^{[0:-m]} \| \pi_\theta \big( \cdot \mid q_k, a_k^{[0:-m]} \big), \quad \text{where} \quad \begin{cases} a_k := \{a \mid (q_k, a) \in \mathcal{C}\} \\ m \sim \mathcal{U}\{0, 1, ..., L\} \end{cases} \tag{3}$$

Here, $L$ is the maximum truncation length, $\mathcal{U}\{0, 1, ..., L\}$ samples a truncation point uniformly from $[0, L]$, $a_k^{[0:-m]}$ truncates the last $m$ tokens of $a_k$, and $\pi_\theta(\cdot|q_k, a_k^{[0:-m]})$ generates a new continuation based on the question and prefix. In this paper, $L$ is either fixed or set dynamically based on the shortest response in a sampling group $G$, denoted as $\ell$, where:

$$\ell = \min\{\text{len}(o_1), \text{len}(o_2), \ldots, \text{len}(o_G)\} \tag{4}$$

**Policy Optimization with Experience Replay**    After completing the sampling generation, POER adopts **Group Relative** estimation of advantage. For a given question-answer pair $(q_k, a_k)$, the behavioral policy $\pi_{\theta_{t-1}}$ samples a group of $G$ individual responses $\{o_i\}_{i=1}^G$ from the model. Then, by normalizing the group rewards $\{R_i\}_{i=1}^G$, the advantage of each response is computed as:

$$\mathcal{J}_{\text{POER}}(\theta_t) = \mathbb{E}_{(q,a)\sim\mathcal{C}^{(t-1)}(Q,A),\{o_i\}_{i=1}^G \sim a_k^{[0:-m]}\|\pi_\theta(\cdot \mid q_k, a_k^{[0:-m]})}$$

$$\frac{1}{G}\sum_{i=1}^G \frac{1}{|o_i|}\sum_{j=1}^{|o_i|}\left\{\min\left[r_{i,j}(\theta_t)\hat{A}_{i,j}, \text{clip}\left(r_{i,j}(\theta_t), 1-\epsilon, 1+\epsilon\right)\hat{A}_{i,j}\right] - \beta D_{\text{KL}}(\pi_{\theta_t}\|\pi_{\text{ref}})\right\} \tag{5}$$

where:

$$r_{i,j}(\theta_t) = \frac{\pi_{\theta_t}(o_{i,j}|q, o_{i,<j})}{\pi_{\theta_{t-1}}(o_{i,j}|q, o_{i,<j})}, \quad \hat{A}_{i,j} = \frac{R_i - \text{mean}(\{R_i\}_{i=1}^G)}{\text{std}(\{R_i\}_{i=1}^G)} \tag{6}$$

Since POER is policy-agnostic, we propose a unified forward reinforcement learning paradigm based on experience replay. We can then write the policy gradient function of POER in a more general form as:

$$\nabla_\theta \mathcal{J}_{\text{POER}}(\theta) = \underbrace{\mathbb{E}_{(q,o)\sim\mathcal{C}}}_{\text{Data Source}}\left(\frac{1}{|o|}\sum_{j=1}^{|o|}\underbrace{\mathcal{G}(q, o, j, \pi_{ref})}_{\text{Gradient Coefficient}}\nabla_\theta \log \pi_\theta(o_j|q, o_{<j})\right) \tag{7}$$

Equation 7 is derived from the standard policy gradient formulation. The above equations indicate that only the sampling stage is affected by POER, while the policy gradient function remains unaltered. As a result, POER exhibits a plug-and-play nature and can be easily integrated into other reinforcement fine-tuning algorithms.

Compared to the traditional GRPO strategy, a previously sampled historical response trajectory is introduced by POER as a constraint into the subsequent sampling process. In this way, the policy space $\pi_\theta$ explored during training is confined. Such a constraint regularizes the gradient descent space during learning, which can be expressed as:

$$Var(\|\nabla_\theta \mathcal{J}_{\text{POER}}\|_2) \leq Var(\|\nabla_\theta \mathcal{J}_{\text{GRPO}}\|_2) \tag{8}$$

In theory, our method enables a more stable training process. We provide detailed mathematical proofs in Appendix B.

### 3.3 CACHE POOL UPDATE

After each gradient update, we adopt the $\varepsilon$-greedy algorithm to update the experience cache by selecting the highest-reward response from the current inference results. Specifically, when the random variable $u \sim \mathcal{U}(0,1)$ satisfies $u \leq \varepsilon$, we select the response with the highest group reward; otherwise, we randomly select a suboptimal response. We formalize the update process as:

$$\mathcal{C}^{(t)} = \begin{cases} \left( \mathcal{C}^{(t-1)} \cup \left\{ (q_k, o_{\mathrm{argmax}\{R_i\}_{i=1}^G}) \right\} \right) \setminus \{(q_k, a_k)\}, & \text{if } u \leq \varepsilon, \\ \left( \mathcal{C}^{(t-1)} \cup \{(q_k, o_{g'})\} \right) \setminus \{(q_k, a_k)\}, & \text{otherwise.} \end{cases} \tag{9}$$

Here, we denote the highest-reward response in the group as $o_{\mathrm{argmax}\{R_i\}_{i=1}^G}$, and we randomly select another candidate response as $o_{g'}$.

### 3.4 LENGTH-AWARE REWARD SHAPING

However, since the same response prefix is shared during the sampling phase, the diversity of responses within the group is reduced compared to the GRPO algorithm. This lower diversity results in more similar reward signals, thereby diminishing the effectiveness of policy gradient estimation. To ensure meaningful gradients, reasonable reward differences are maintained within the group, even when all responses are correct.

A **Length-Aware Reward Shaping** method is proposed to address this issue. This method is based on the assumption that: *For the same question, a reasoning path that reaches the correct answer more concisely should be rewarded with a higher value.* Specifically, for each response $o_i$ in the group, its length-aware reward $R(s_i)$ is computed as:

$$R(s_i) = \mathrm{clip}\left( \frac{r(s_i)}{1 + e^{-\alpha(\ell_{\mathrm{ref}} - \mathrm{len}(o_i))}}, \ m, \ \mathcal{M} \right) \tag{10}$$

Here, $r(s_i)$ is the original reward, and $\ell_{\mathrm{ref}}$ is the average length of group $G$, defined as $\ell_{\mathrm{ref}} = \frac{1}{|G|} \sum_{i=1}^{|G|} \mathrm{len}(o_i)$. The parameter $\alpha > 0$ controls the sensitivity of the reward to length differences. $m$ and $\mathcal{M}$ are the lower and upper bounds for reward clipping to avoid extremely large or small values. $\mathrm{clip}(\cdot, m, \mathcal{M})$ denotes restricting a value within the interval $[m, \mathcal{M}]$.

We then iterate the above steps in Sections 3.2 and 3.3 until a predefined stopping step $T$ is reached.

Through mathematical derivation, we demonstrate that length-aware rewards are better suited for the POER algorithm; the two can complement each other, and when the guiding path is within a certain threshold, they can enable the model to achieve greater performance gains. In contrast, the GRPO algorithm, lacking an initial fixed guiding path, results in high variance for length-aware rewards, making it difficult to accurately estimate the true effective policy gradient, and is therefore not suitable for using length-aware rewards. Detailed proof is provided in Appendix C.

## 4 EXPERIMENTS

### 4.1 EXPERIMENTAL SETUP

**Base Models**  To demonstrate the effectiveness and generality of POER, we evaluate it on two open-source inference models with 1.5B and 7B parameters, namely **Deepseek-r1-qwen-distill-1.5b** and **Deepseek-r1-qwen-distill-7b** (DeepSeek-AI et al., 2025; Bai et al., 2023). Notably, we skip the supervised fine-tuning (SFT) phase, which is usually a prerequisite for reinforcement learning to enhance performance (Chu et al., 2025), as the selected models have already undergone this stage (DeepSeek-AI et al., 2025).

**Evaluation and Datasets**  We evaluate the models on six standard reasoning evaluation datasets: aime25(math ai, b), aime24(math ai, a), math500(Hendrycks et al., 2021), amc23(math ai, c), minerva(Lewkowycz et al., 2022) and olympicbench(He et al., 2024). To ensure fairness, all evaluations use the lighteval(Habib et al., 2023) toolkit.

**Implementation Details** During training, we use 7k samples from the open-rs dataset (Dang & Ngo, 2025) with a global batch size of 576 for 4 epochs. Experiments are run on a single H20 machine with 8×H20 96G GPUs. We generate 6 samples per prompt, set the temperature to 0.7, and fix the maximum generation length at 4096. For the length-aware reward, we use $m = 0.5$, $M = 1$, and $\alpha = 0.01$. All models are fully fine-tuned. Due to time constraints, only zero-shot performance is averaged over three runs; all other ablation experiments are run once.

## 4.2 ZERO-SHOT PERFORMANCE

We set the maximum truncation length $L$ for each group to half of the minimum response length $\ell$. Then, we train the DeepSeek-R1-Qwen-1.5B and DeepSeek-R1-Qwen-7B models for four epochs using the original GRPO algorithm, the DAPO algorithm, as well as their POER variants, with a batch size of 576 and a maximum generation length of 4096 tokens. To ensure that the experimental results are not caused by randomness, we repeat the training three times for each experiment. We then compare their mean accuracy on the designated evaluation datasets.

Table 1: Performance of the POER algorithm on test datasets. Arrows indicate performance changes relative to the base model: ↑ indicates improvement, ↓ indicates decline. w/ R means length-aware reward is used, w/o R means length-aware reward is not used. +POER shows the effect of applying the POER algorithm on top of the above method.

| Model | AIME25 | AIME24 | MATH500 | AMC23 | Minerva | OlyB | Avg |
|---|---|---|---|---|---|---|---|
| **1.5B Models** | | | | | | | |
| DeepSeek-R1-Qwen-1.5B | 16.7 | 28.8 | 82.2 | 62.9 | 26.5 | 43.3 | 43.4 |
| + GRPO(w/o R) | 24.4↑ | 31.1↑ | 85.7↑ | 72.5↑ | 29.8↑ | 51.3↑ | 49.1↑ |
| + **POER** | 24.4↑ | 25.6↓ | 84.3↑ | 69.2↑ | 29.5↑ | 51.7↑ | 47.5↑ |
| + GRPO(w/ R) | 22.2↑ | 32.2↑ | 83.8↑ | 70.8↑ | 27.5↑ | 50.5↑ | 47.8↑ |
| + **POER** | 24.4↑ | 35.6↑ | 85.3↑ | 83.3↑ | 29.8↑ | 51.8↑ | 51.7↑ |
| + DAPO(w/o R) | 30.0↑ | 24.4↓ | 86.2↑ | 84.2↑ | 29.7↑ | 52.7↑ | 51.2↑ |
| + **POER** | 28.9↑ | 24.4↓ | 86.0↑ | 84.5↑ | 29.3↑ | 52.1↑ | 50.9↑ |
| + DAPO(w/ R) | 26.7↑ | 30.0↑ | 85.0↑ | 84.1↑ | 29.7↑ | 51.1↑ | 50.2↑ |
| + **POER** | 32.2↑ | 30.0↑ | 86.2↑ | 86.1↑ | 29.1↑ | 52.3↑ | 52.7↑ |
| **7B Models** | | | | | | | |
| DeepSeek-R1-Qwen-7B | 43.3 | 55.5 | 92.8 | 90.0 | 44.5 | 67.4 | 65.6 |
| + GRPO(w/o R) | 43.3 | 53.3↓ | 95.0↑ | 90.0 | 44.5 | 67.2↓ | 65.6 |
| + **POER** | 43.3 | 46.6↓ | 92.5↓ | 89.2↑ | 42.3↓ | 67.7↑ | 63.6↓ |
| + GRPO(w/ R) | 40.0↓ | 48.9↓ | 95.0↑ | 88.3↑ | 43.5↓ | 66.0↓ | 63.6↓ |
| + **POER** | 50.0↑ | 61.1↑ | 94.2↑ | 90.8↑ | 43.7↓ | 67.3↓ | 67.8↑ |
| + DAPO(w/o R) | 43.3 | 53.3↓ | 94.6↑ | 90.2↑ | 45.1↑ | 67.7↑ | 65.7↑ |
| + **POER** | 46.7↑ | 52.2↓ | 94.2↑ | 91.2↑ | 42.7↓ | 64.9↓ | 65.3↓ |
| + DAPO(w/ R) | 42.2↓ | 56.7↑ | 93.2↑ | 91.8↑ | 44.6↑ | 64.5↓ | 65.5↓ |
| + **POER** | 46.7↑ | 54.5↓ | 94.8↑ | 95.2↑ | 43.1↓ | 64.5↓ | 66.5↑ |

As shown in Table 1, as mentioned in the Method section, length-aware rewards complement the POER algorithm. Incorporating group-wise length-aware rewards enables POER to achieve higher accuracy on test benchmarks than GRPO and DAPO for both the 1.5B and 7B model sizes. Without group-wise length-aware rewards, POER may experience some performance degradation; therefore, when using POER for accelerated training, it is recommended to include group-wise length-aware rewards to enhance performance.

## 4.3 TRAINING TIME OVERHEAD

To investigate the training time overhead of GRPO and POER, each experiment is conducted on a machine with 8 H20 GPUs, using only a single GPU for sampling during the training phase. It should be noted that POER introduces additional inference overhead during the cache initialization phase, where parallel inference is performed across all GPUs using the vllm framework. When the dataset size is 7k and the parallel batch size is 256, this phase takes approximately 20 minutes. Our experiments reveal that the primary factors affecting the relative training speed between POER and GRPO are the number of group samples $G$ and the maximum truncation length $L$, while the impact of batch size is relatively minor.

Table 2: The average number of tokens generated per sample with the POER method

| $L$ | 1.5B Model | 7B Model |
|-----|------------|----------|
| 300 | 145.88 | 147.06 |
| 500 | 158.41 | 168.17 |
| 800 | 382.20 | 397.89 |
| GRPO | 2689.51 | 2457.91 |

Table 3: Training time of POER and GRPO under 4 epochs with $L = 800$, h represents hours

| Method | 1.5B Model | 7B Model |
|--------|------------|----------|
| GRPO | 77.28 h | 84.53 h |
| +POER | 8.37 h | 23.50 h |

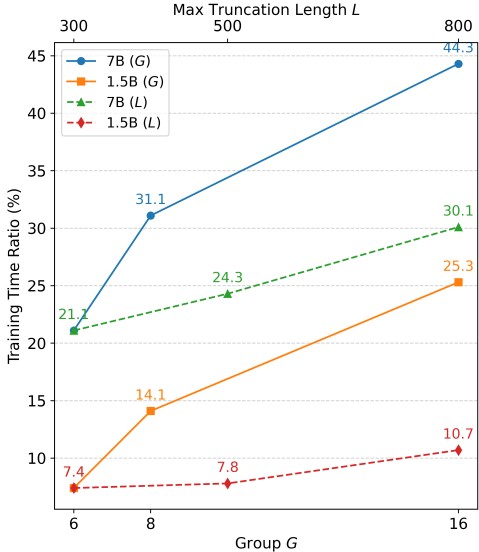

Figure 3: The impact of maximum truncation length $L$ and group size $G$ on the acceleration ratio of POER under the 1.5B and 7B settings. Due to the memory limitation of a single machine, the maximum generation length for 7B $(G)$ is 2048, while for all other cases it is 4096.

We study training speed for both 1.5B and 7B models. With maximum truncation length fixed at $L = 300$, we set per-GPU batch sizes of 2 (1.5B) and 1 (7B), and evaluate group sizes $G = 6, 8, 16$. As shown in Figure 3, smaller $G$ yields greater acceleration for POER, reducing training time to 7.4% of GRPO for 1.5B and 21.1% for 7B. We also study the effect of $L$ with $G = 6$. Figure 3 shows that larger $L$ increases POER's relative training time, with a smaller rise for the 7B model than for the 1.5B model.

The actual training speed is affected by many factors, so we propose a fairer comparison: using the average tokens generated per sample. Since prefill is much faster than decoding, more tokens in prefill lead to shorter decoding time. As shown in Table 2, under the original GRPO algorithm, each sample requires an average of 2689.51 tokens and 2457.91 tokens for the 1.5B and 7B models, respectively. In contrast, with the POER algorithm, the number can be reduced to as low as 145.88 tokens and 147.06 tokens. From the perspective of the decode stage, the time overhead of POER is only about 5% of that of GRPO. Table 3 presents the detailed training time overhead of the original GRPO and POER algorithms over 4 epochs.

It is worth noting that the average number of tokens generated by the GRPO algorithm for the 1.5B model is higher than that for the 7B model. However, when constrained by POER, the number of tokens generated is lower. This is because the POER algorithm preserves shorter correct answers, and the exploration capability of the 1.5B model, once guided, is weaker compared to that of the 7B model, leading to this phenomenon.

### 4.4 STABILITY ANALYSIS

Traditional RL methods like GRPO and PPO are unstable in multi-step training: performance often degrades with more iterations, and response length tends to shorten. Consequently, GRPO fine-tuning usually limits iteration numbers to prevent deterioration, with accuracy and response length used to measure model degradation (DeepSeek-AI et al., 2025). POER addresses this by using a cache pool mechanism, and we conduct comparative experiments to quantify its improved training stability over GRPO.

During the training process, we use a batch size of 18 to train the 7B and 1.5B models for four epochs, and monitor changes in response length and model performance, as shown in Figure 4 and 5. In this

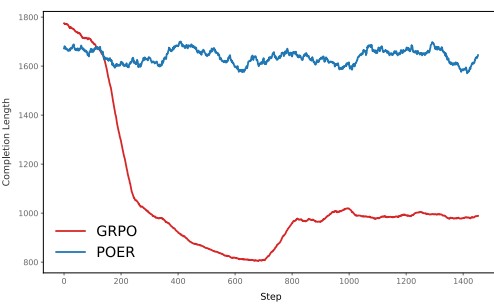 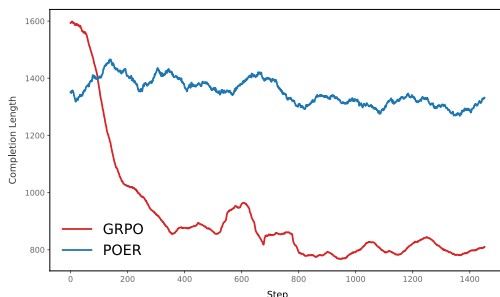

Figure 4: Response Length of GRPO and POER on a 1.5B Model as a Function of Training Step

Figure 5: Response Length of GRPO and POER on a 7B Model as a Function of Training Step

multi-step iterative training setup, GRPO experiences a collapse in response length around the 200th iteration, while POER maintains stable response lengths throughout the process. On the other hand, as shown in Table 4, the model performance after training with GRPO deteriorated, especially for the 1.5B model, where accuracy dropped by 8.6%. In contrast, POER results in a 5.4% improvement in accuracy.

Table 4: Performance of GRPO and POER on Evaluation Datasets in Multi-Step Iteration Scenarios

| Model | AIME25 | AIME24 | MATH500 | AMC23 | Minerva | OlyB | Avg |
|---|---|---|---|---|---|---|---|
| DeepSeek-R1-Qwen-7B | 43.3 | 55.5 | 92.8 | 90.0 | 44.5 | 67.4 | 65.6 |
| + GRPO(w/o R) | 40.0↓ | 50.0↓ | 94.2↑ | 90.0 | 41.2↓ | 66.7↓ | 63.7↓ |
| **+ POER** | 46.6↑ | 56.7↑ | 92.8 | 90.0 | 41.4↓ | 66.1↓ | 65.6 |
| DeepSeek-R1-Qwen-1.5B | 16.7 | 28.8 | 82.2 | 62.9 | 26.5 | 43.3 | 43.4 |
| + GRPO(w/o R) | 10.0↓ | 10.0↓ | 67.0↓ | 45.0↓ | 20.6↓ | 31.4↓ | 34.8↓ |
| **+ POER** | 20.0↑ | 36.7↑ | 82.8↑ | 72.5↑ | 29.4↑ | 51.5↑ | 48.8↑ |

Beyond the instability from reward sparsity, GRPO suffers from strong locality due to its inter-group comparison strategy, limiting performance improvements. POER mitigates this by introducing an experience cache, using an external cached policy $\pi_{\mathcal{C}}$ to approximate the main policy $\pi_\theta$ during updates. This provides a global context, enhances training stability, and allows POER to maintain consistent performance over long iterations.

Table 5: Impact of $\alpha$ and $L$ on validation accuracy (%) of DeepSeek-R1-Qwen-1.5B

| $L$ | $\alpha$ | AIME25 | AIME24 | MATH500 | AMC23 | Minerva | OlyB | Avg |
|---|---|---|---|---|---|---|---|---|
| | 0 | 26.7 | 33.3 | 82.8 | 70.0 | 27.9 | 52.4 | 48.9 |
| 300 | 0.01 | 26.7 | 33.3 | 85.2 | 77.5 | 29.4 | 52.0 | 50.7 |
| | 0.1 | 23.3 | 36.7 | 85.8 | 75.0 | 32.4 | 51.0 | 50.7 |
| | 1 | 23.3 | 36.7 | 83.2 | 72.5 | 31.6 | 53.2 | 50.1 |
| | 0 | 30.0 | 33.3 | 84.4 | 70.0 | 29.0 | 50.5 | 49.5 |
| $0.5\ell$ | 0.01 | 23.3 | 16.7 | 86.4 | 67.5 | 32.0 | 55.3 | 46.9 |
| | 0.1 | 36.7 | 33.3 | 85.2 | 75.0 | 27.9 | 52.7 | 51.8 |
| | 1 | 30.0 | 30.0 | 82.6 | 60.0 | 31.6 | 51.6 | 47.6 |
| | 0 | 33.3 | 26.7 | 84.6 | 75.0 | 27.8 | 53.3 | 50.1 |
| $\ell$ | 0.01 | 36.7 | 30.0 | 84.0 | 70.0 | 31.6 | 52.3 | 50.7 |
| | 0.1 | 30.0 | 36.7 | 84.4 | 70.0 | 28.3 | 53.5 | 50.4 |
| | 1 | 26.7 | 26.7 | 85.4 | 65.0 | 30.5 | 52.3 | 47.8 |

## 5 MORE ANALYSIS

**Impact of Maximum Truncation Length and** $\alpha$ Intuitively, the maximum truncation length $L$ and $\alpha$ are not independent factors. To study their effect on training, we train the model with combinations of $L = 300, 0.5\ell, \ell$ and $\alpha = 0, 0.01, 0.1, 1$. Notably, when $\alpha = 0$, the intra-group length-aware reward is disabled, so all correct reasoning paths receive the same reward.

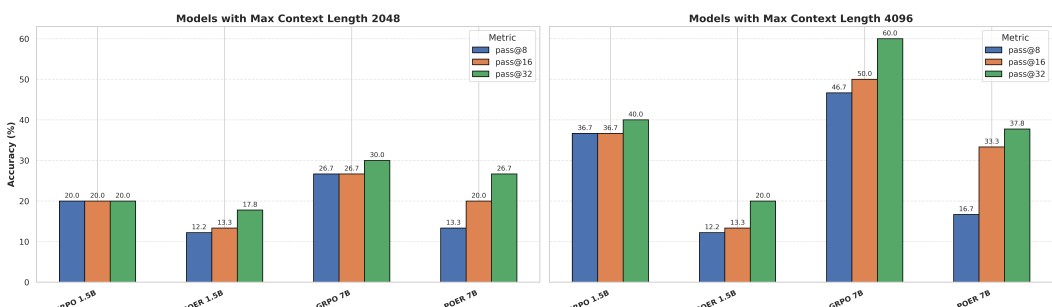

Figure 6: Pass@N performance of the GRPO and POER algorithms on the AIME24 dataset with the maximum truncation length set to 300. The left figure shows the case with a maximum generation length of 2048, and the right figure shows the case with a maximum generation length of 4096

The DeepSeek-R1-Qwen-1.5B model is trained for two epochs with a batch size of 336 to amplify differences in training outcomes for easier observation. The evaluation results are shown in Table 5: under a fixed $L$, performance first improves as $\alpha$ increases and then declines.

**Effect of max_length on Exploration Capability**  To investigate the impact of `max_length` settings on the model's initial exploration ability, we set the maximum truncate length of POER to 300 and examined the performance of the 1.5B and 7B models on the AIME24 dataset under two settings: `max_length = 2048` and `max_length = 4096`. As shown in Figure 6, POER demonstrates lower exploration ability compared to GRPO, and the gap between the two methods gradually widens as `max_length` increases. This result also indicates that POER exhibits a certain disadvantage in exploration ability during the early iterations.

**Impact of Cache Pool Update Strategy on Model's Pass@N Performance**  To study the effect of training epochs on exploration, we evaluate the 1.5B and 7B models on AIME24 with `max_length` set to 2048 and 4096 under `epoch = 1,2`. As shown in Figure 7, updating the cache pool over epochs enables the model to explore more diverse and higher-quality solution paths, steadily improving performance to match or surpass GRPO. Overall, while POER reduces raw exploration, the experience cache and epsilon-greedy strategy guide the model toward higher-quality paths.

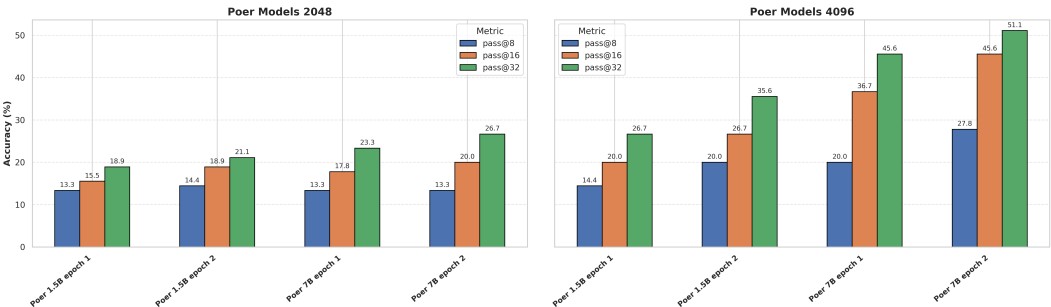

Figure 7: PassN performance of the GRPO and POER algorithms on the AIME24 dataset with the maximum truncation length set to 300 and epochs set to 1 and 2. The left figure shows the case with a maximum generation length of 2048, and the right figure shows the case with a maximum generation length of 4096

## 6 CONCLUSION

In this paper, we present POER, a plug-and-play algorithm designed to optimize the reinforcement fine-tuning of large models. POER aims to enhance the fine-tuning phase of large language models by introducing an experience replay mechanism. This mechanism allows the model to learn from previously collected high-quality responses during generation. POER significantly reduces model training time while improving the fine-tuned model's performance and enhancing stability during the reinforcement fine-tuning phase.

## THE USAGE OF LLM

In this work, we use LLMs to polish the paper, generate materials for framework diagrams, and retrieve related work.

## ETHICS STATEMENT

This study does not involve any personal data, sensitive information, or high-risk application scenarios. No ethically controversial datasets or models were used. All experimental data are standard benchmark datasets that are publicly available, and the sole purpose of this research is to advance the development of reinforcement fine-tuning algorithm. Therefore, we believe this work does not pose any significant ethical risks.

## REPRODUCIBILITY STATEMENT

To ensure the reproducibility of our experiments, we have provided the complete implementation code in the supplementary materials. All technical details, including the evaluation benchmarks, baseline methods, and training hyperparameter settings used in this work, can be found in Section 4.1.

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

## A    PSEUDO CODE FOR POER TRAINING PROCESS

The pseudo code of *Policy Optimization with Experience Replay*(POER) in the training process is as follows.

---
**Algorithm 1:** Compact POER Training

---
**Input:** Dataset $\mathcal{D}$, model $\pi_{\theta_0}$, cache $\mathcal{C}^{(0)}$, params $\eta, \beta, \epsilon, G$
**Output:** Model $\pi_{\theta_T}$, cache $\mathcal{C}^{(T)}$
**Initialize Cache:**
$\mathcal{C}^{(0)} = \emptyset$
**for** $q_k \in \mathcal{D}$ **do**
    $\lfloor\ \mathcal{C}^{(0)} \leftarrow \mathcal{C}^{(0)} \cup \{(q_k, \pi_{\theta_0}(\cdot|q_k))\}$
**for** $t \leftarrow 1$ **to** $T$ **do**
    **Rollout:**
    **for** $q_k \in \mathcal{D}$ **do**
        $a_k \leftarrow \{a \mid (q_k, a) \in \mathcal{C}^{(t-1)}\}$
        **for** $g \leftarrow 1$ **to** $G$ **do**
            $\tilde{a}_i^{(g)} \leftarrow \text{Concat}(a_k^{[0:-m]}, \pi_{\theta_{t-1}}(\cdot|q_k, a_k^{[0:-m]}))$
            $R_i^{(g)} \leftarrow r(q_k, \tilde{a}_i^{(g)})$
    **Optimize:**
    $\theta_t \leftarrow \theta_{t-1} + \eta \nabla_\theta J_{\text{GRPO-Cache}}$
    **Update Cache:**
    **for** $q_k \in \mathcal{D}$ **do**
        **if** $u \leq \epsilon$ **then**
            $\mathcal{C}^{(t-1)} \cup \left\{ (q_k, o_{\arg\max\{R_i\}_{i=1}^G}) \right\} \setminus \{(q_k, a_k)\}$
        **else**
            $\mathcal{C}^{(t)} \leftarrow \mathcal{C}^{(t-1)} \cup \{(q_k, o_{g'})\} \setminus (q_k, a_k)$ ;           // $g' \sim \mathcal{U}(1, G)$

---

## B    PROOF OF POER GRADIENT STABILITY

**Policy gradient estimation.**    The reasoning process of traditional GRPO and *Policy Optimization with Experience Replay*(POER) can be expressed as $\pi(\cdot|q_k)$ and $\pi(\cdot|(q_k, a_k))$, where $q_k \in C_{raw}$, is a sample in raw training dataset $C_{raw}$ and corresponding $(q_k, a_k) \in C$, is one example in training replay buffer $C$, which updates during the training process.

For any sample $q_k$, it holds that $(q_k) \subset (q_k, a_k)$. Hence, for the response space of an arbitrary policy model, the total variance can give

$$\text{Var}(\cdot \mid q_k) = \mathbb{E}_{(q_k, a_k)|q_k}\Big[\text{Var}\big(\cdot \mid (q_k, a_k)\big)\Big] + \text{Var}_{(q_k, a_k)|q_k}\Big(\mathbb{E}\big[\cdot \mid (q_k, a_k)\big]\Big). \tag{11}$$

Because the second term on the right-hand side is non-negative, i.e. $\text{Var}_{(q_k, a_k)|q_k}\left(\mathbb{E}[\cdot|(q_k, a_k)]\right) \geq 0$, we obtain

$$\text{Var}(\cdot \mid q_k) \ \geq \ \mathbb{E}_{(q_k, a_k)|q_k}\Big[\text{Var}\big(\cdot \mid (q_k, a_k)\big)\Big]. \tag{12}$$

Treating $(q_k, a_k)$ as an augmentation of $q_k$ allows this inequality to simplify to

$$\text{Var}(\cdot \mid q_k) \ \geq \ \text{Var}\big(\cdot \mid (q_k, a_k)\big). \tag{13}$$

In the policy space, equation 13 becomes

$$\text{Var}\big(\pi_\theta(a_g \mid q_k)\big) \ \geq \ \text{Var}\big(\pi_\theta(a_g \mid (q_k, a_k))\big). \tag{14}$$

Assume there exists a parameter vector $\theta_0$ such that the policy can be locally approximated by the first-order expansion

$$\pi_\theta = \pi_{\theta_0} + \nabla\pi^\top(\theta_0)\big(\theta - \theta_0\big). \tag{15}$$

The variance of $\pi_\theta$ in a neighbourhood of $\theta_0$ can then be estimated as

$$\sigma^2\big(\pi_\theta\big) \approx \nabla\pi_{\theta_0}^\top \Sigma_\theta \nabla\pi_{\theta_0}, \tag{16}$$

where $\Sigma_\theta$ denotes the covariance matrix of the parameter estimates.

Throughout training, the realizations of $\pi_\theta$ can be treated as i.i.d. random variables. As the sample size $n \to \infty$, the empirical mean and variance converge to

$$\mu(\pi_\theta) = \mathbb{E}[\pi_\theta] = \frac{1}{n}\sum_{i=1}^{n}(\pi_\theta)_i, \qquad \mathrm{Var}(\pi_\theta) = \mathbb{E}[\pi_\theta^2] - \mu(\pi_\theta)^2 \approx \frac{n}{n-1}\,\nabla\pi_{\theta_0}^\top\Sigma_\theta\,\nabla\pi_{\theta_0}.$$

On account of $\mathbb{E}[\pi_\theta^2] = \mu(\pi_\theta)^2$, while $\Sigma_\theta = I$ also holds, then

$$\sigma^2(\pi_\theta) = \mathrm{Var}(\pi_\theta) = \nabla\pi_{\theta_0}^\top\nabla\pi_{\theta_0} = \left\|\nabla\pi_{\theta_0}\right\|_2^2. \tag{17}$$

Combining the above with equation 13 yields the policy space gradient estimation.

$$\left\|\nabla\pi_\theta\big(a_g \mid q_k\big)\right\|_2 \;\geq\; \left\|\nabla\pi_\theta\big(a_g \mid (q_k, a_k)\big)\right\|_2. \tag{18}$$

This establishes that conditioning on the augmented information $(q_k, a_k)$ strictly reduces—or at worst preserves—the magnitude of the policy-gradient variance.

Let's review the GRPO update policy. At training step $t$, the optimisation target of *Generative Reinforcement Policy Optimisation* (GRPO) can be written as

$$\mathcal{J}_{\mathrm{GRPO}}(\theta_t) = \mathbb{E}_{(q,a)\sim\mathcal{C}^{(t-1)}(Q,A),\{o_i\}_{i=1}^{G}\sim\pi_{\theta_{t-1}}(\cdot|q)}$$

$$\left[\frac{1}{G}\sum_{i=1}^{G}\frac{1}{|o_i|}\sum_{j=1}^{|o_i|}\min\left(r_{i,j}(\theta_t)\hat{A}_{i,j}, \mathrm{clip}\Big(r_{i,j}(\theta_t), 1-\epsilon, 1+\epsilon\Big)\hat{A}_{i,j} - \beta D_{\mathrm{KL}}(\pi_{\theta_t}\|\pi_{\mathrm{ref}})\right)\right] \tag{19}$$

Here

$$r_{i,j}(\theta_t) = \frac{\pi_{\theta_t}(o_{i,j} \mid q)}{\pi_{\theta_{t-1}}(o_{i,j} \mid q)}, \qquad \hat{A}_{i,j} = \frac{R_i - \mathrm{mean}\big(\{R_i\}_{i=1}^{G}\big)}{\mathrm{std}\big(\{R_i\}_{i=1}^{G}\big)}.$$

Relative to GRPO, POER only changes the policy ratio by conditioning on the prefix $o_{i,<j}$:

$$r_{i,j}^{\mathrm{POER}}(\theta_t) = \frac{\pi_{\theta_t}(o_{i,j} \mid q,\, o_{i,<j})}{\pi_{\theta_{t-1}}(o_{i,j} \mid q,\, o_{i,<j})}. \tag{20}$$

Because the `clip` operation truncates high-error updates, both algorithms behave identically whenever clipping is activated.

**Gradient of the optimization policy.** For the plain GRPO ratio one obtains

$$\nabla_\theta\, r_{i,j}(\theta_t) = \frac{\nabla_\theta\pi_{\theta_t}(o_{i,j} \mid q)}{\pi_{\theta_{t-1}}(o_{i,j} \mid q)} = \frac{\pi_{\theta_t}(o_{i,j} \mid q)}{\pi_{\theta_{t-1}}(o_{i,j} \mid q)}\,\nabla_\theta\log\pi_{\theta_t}(o_{i,j} \mid q). \tag{21}$$

The Kullback–Leibler divergence with respect to a frozen reference policy $\pi_{\mathrm{ref}}$ satisfies

$$\nabla_\theta D_{\mathrm{KL}}\big(\pi_{\theta_t} \| \pi_{\mathrm{ref}}\big) = \nabla_\theta\mathbb{E}_{\pi_{\theta_t}}\left[\log\pi_{\theta_t} - \log\pi_{\mathrm{ref}}\right]$$

$$= \mathbb{E}_{\pi_{\theta_t}}\left[\nabla_\theta\log\pi_{\theta_t} + \big(\log\pi_{\theta_t} - \log\pi_{\mathrm{ref}}\big)\nabla_\theta\log\pi_{\theta_t}\right] \tag{22}$$

$$= \mathbb{E}_{\pi_{\theta_t}}\left[\nabla_\theta\log\pi_{\theta_t}\big(\log\tfrac{\pi_{\theta_t}}{\pi_{\mathrm{ref}}} + 1\big)\right]$$

As $\mathbb{E}_{\pi_{\theta_t}}[\nabla_\theta\log\pi_{\theta_t}] = 0$ by normalisation, the expression simplifies to

$$\nabla_\theta D_{\mathrm{KL}} = \mathbb{E}_{\pi_\theta}\left[\nabla_\theta\log\pi_\theta(o|s)\cdot\log\pi_\theta(o|s)\right]$$

$$= \sum_{\pi_\theta}\pi_\theta\nabla_\theta\log\pi_\theta(o|s)\cdot\log\pi_\theta(o|s) \tag{23}$$

$$= \frac{1}{|o_i|}\sum_{t=1}^{|o_i|}\pi_\theta\nabla_\theta\log\pi_\theta(o_i|s)\cdot\log\pi_\theta(o_i|s)$$

**Resulting policy gradient.** Aggregating the intra-group updates yields the estimator

$$
\nabla_\theta \mathcal{J}_{\mathrm{GRPO}} = \mathbb{E}_{q,\{o_i\}} \left[ \frac{1}{G} \sum_{i=1}^{G} \frac{1}{|o_i|} \sum_{j=1}^{|o_i|} \left( \frac{\hat{A}_{i,j}}{\pi_{\theta_{t-1}}(o|q)} - \beta \log \pi_{\theta_t}(o \mid q) \right) \pi_{\theta_t}(o \mid q) \, \nabla_\theta \log \pi_{\theta_t}(o \mid q) \right]
$$

$$
= \mathbb{E}_{q,\{o_i\}} \left[ \frac{1}{G} \sum_{i=1}^{G} \frac{1}{|o_i|} \sum_{j=1}^{|o_i|} \left( \frac{\hat{A}_{i,j}}{\pi_{\theta_{t-1}}(o|q)} - \beta \log \pi_{\theta_t}(o \mid q) \right) \nabla_\theta \pi_{\theta_t}(o \mid q) \right].
$$

(24)

The second line follows by noting that $\pi_{\theta_t} \nabla_\theta \log \pi_{\theta_t} = \nabla_\theta \pi_{\theta_t}$. Equation above provides the final form of the GRPO gradient used for parameter updates at step $t$.

**Without consideration of KL divergence.** If the KL–divergence term is temporarily ignored, the GRPO gradient estimator reduces to

$$
\nabla_\theta \mathcal{J}_{\mathrm{GRPO}} = \mathbb{E}_{q,\{o_i\}} \left[ \frac{1}{G} \sum_{i=1}^{G} \frac{1}{|o_i|} \sum_{t=1}^{|o_i|} \left( \frac{\hat{A}_{i,t}}{\pi_{\theta_{t-1}}} \right) \nabla \pi_\theta(a_g \mid q_k) \right].
$$

(25)

Because of equation equation 18,

$$
\| \nabla_\theta \mathcal{J}_{\mathrm{GRPO}} \|_2 \geq \| \nabla_\theta \mathcal{J}_{\mathrm{POER}} \|_2.
$$

(26)

**Including the KL divergence.** At the initial step ($t = 0$) both algorithms share the same reference policy, hence

$$
\nabla_\theta \mathcal{J}_{\mathrm{GRPO}} = \nabla_\theta \mathcal{J}_{\mathrm{POER}}.
$$

(27)

For the first update ($t = 1$) equation 18 implies

$$
\| \nabla \pi_{\theta_1}(a_g \mid q_k) \|_2 \geq \| \nabla \pi_{\theta_1}(a_g \mid q_k, a_k) \|_2.
$$

(28)

Here we record $\nabla \pi_{\theta_1}(a_g \mid q_k, a_k)$ as $\nabla \pi'_{\theta_1}$. Consequently, the difference of the two policy gradients becomes

$$
\Delta_1 = \| \nabla_\theta \mathcal{J}_{\mathrm{GRPO}} \|_2 - \| \nabla_\theta \mathcal{J}_{\mathrm{POER}} \|_2
$$

$$
= \mathbb{E}_{q,\{o_i\}} \left[ \frac{1}{G} \sum_{i=1}^{G} \frac{1}{|o_i|} \sum_{t=1}^{|o_i|} \left( \| (\frac{\hat{A}_{i,1}}{\pi_{\theta_0}} - \beta \log \pi_{\theta_1}) \nabla \pi_{\theta_1} \|_2 - \| (\frac{\hat{A}'_{i,1}}{\pi_{\theta_0}} - \beta \log \pi'_{\theta_1}) \nabla \pi'_{\theta_1} \|_2 \right) \right].
$$

(29)

Let $\delta := \nabla_\theta \pi_{\theta_1}(i) - \nabla_\theta \pi'_{\theta_1}(i) \geq 0$. For one random dimension the mean-value theorem yields

$$
\pi_{\theta_1}(i) \log \pi_{\theta_1}(i) - \pi'_{\theta_1}(i) \log \pi'_{\theta_1}(i) = \frac{\partial (\pi \log \pi)}{\partial \pi} \Big|_{\pi = \zeta} (\pi_{\theta_1}(i) - \pi'_{\theta_1}(i)), \quad \zeta \in [\pi'_{\theta_1}, \pi_{\theta_1}] \subset [0, 1).
$$

(30)

Taking the directional derivative with respect to $\theta$ gives

$$
(1 + \log \pi_{\theta_1}(i)) \nabla \pi_{\theta_1}(i) - (1 + \log \pi'_{\theta_1}(i)) \nabla \pi'_{\theta_1}(i) = \frac{\partial \pi_\theta \log \pi_\theta}{\partial \pi_\theta} \Big|_{\pi_\theta = \zeta} (\nabla \pi_{\theta_1}(i) - \nabla \pi'_{\theta_1}(i))
$$

$$
= \frac{\partial \pi_\theta \log \pi_\theta}{\partial \pi_\theta} \Big|_{\pi_\theta = \zeta} \delta
$$

(31)

Hence

$$
\log \pi_{\theta_1}(i) \, \nabla \pi_{\theta_1}(i) - \log \pi'_{\theta_1}(i) \, \nabla \pi'_{\theta_1}(i) = \left( \frac{\partial (\pi \log \pi)}{\partial \pi} \Big|_{\pi = \zeta} - 1 \right) \delta = (\log \zeta) \delta \leq 0,
$$

(32)

because $\log \zeta < 0$.

Extending this argument component-wise to the full parameter vector shows

$$
\log \pi_{\theta_1} \nabla \pi_{\theta_1} \preceq \log \pi'_{\theta_1} \nabla \pi'_{\theta_1},
$$

(33)

and therefore

$$\| - \beta \log \pi_{\theta_1} \nabla \pi_{\theta_1} \|_2 \geq \| - \beta \log \pi'_{\theta_1} \nabla \pi'_{\theta_1} \|_2, \tag{34}$$

We have thus established

$$\| \nabla_{\theta_1} \mathcal{J}_{\mathrm{GRPO}} \|_2 \ \geq \ \| \nabla_{\theta_1} \mathcal{J}_{\mathrm{POER}} \|_2. \tag{35}$$

Let the generic update rule be

$$\theta_i = \theta_{i-1} + \eta \, \nabla_\theta \mathcal{J}. \tag{36}$$

Then

$$\frac{\nabla \pi_{\theta_i}}{\pi_{\theta_{i-1}}} = \frac{\nabla \big(\pi_{\theta_{i-1}} + \eta \, \nabla \pi_{\theta_{i-1}} \nabla_{\theta_{i-1}} \mathcal{J}\big)}{\pi_{\theta_{i-1}}} \ \geq \ \frac{\nabla \pi_{\theta_{i-1}}}{\pi_{\theta_{i-1}}} = \nabla \log \pi_{\theta_{i-1}}, \tag{37}$$

i.e. each step re-enters the original policy-gradient (PG) regime. Using equation 35 one obtains for every $i \geq 1$

$$\frac{\nabla \pi_{\theta_i}}{\pi_{\theta_{i-1}}} - \frac{\nabla \pi'_{\theta_i}}{\pi'_{\theta_{i-1}}} \ \geq \ \frac{\nabla \pi_{\theta_{i-1}}}{\pi_{\theta_{i-1}}} - \frac{\nabla \pi'_{\theta_{i-1}}}{\pi'_{\theta_{i-1}}} \ \geq \ \nabla \log \frac{\pi_{\theta_{i-1}}}{\pi'_{\theta_{i-1}}}. \tag{38}$$

By induction this yields the general relation

$$\| \nabla_\theta \mathcal{J}_{\mathrm{GRPO}} \|_2 \ \geq \ \| \nabla_\theta \mathcal{J}_{\mathrm{POER}} \|_2 \qquad \text{for all optimisation steps.} \tag{39}$$

Equation equation 39 completes the proof that, under identical hyper-parameters, GRPO provides gradient updates at least as large as those of POER, both without and with the KL divergence. Meanwhile, since both of them follow a normal distribution with zero mean, it follows that:

$$Var(\| \nabla_\theta \mathcal{J}_{\mathrm{POER}} \|_2) \leq Var(\| \nabla_\theta \mathcal{J}_{\mathrm{GRPO}} \|_2) \tag{40}$$

## C THEOREM

**Preliminary.** Let $q$ denote the prompt, $o$ a sampled response with length $\ell = \mathrm{len}(o)$, and let the group-wise reference length be $\ell_{\mathrm{ref}} = \frac{1}{G} \sum_{i=1}^G \mathrm{len}(o_i)$. Write $\Delta \ell = \ell - \ell_{\mathrm{ref}}$ and fix a window $|\Delta \ell| \leq \tau$. For a sensitivity parameter $\alpha > 0$ define the length weight $s_\alpha(\ell) = \sigma\big(\alpha(\ell_{\mathrm{ref}} - \ell)\big) = (1 + e^{-\alpha(\ell_{\mathrm{ref}} - \ell)})^{-1} \in (0, 1)$ and the shaped reward $R_\alpha = \mathrm{clip}\big(s_\alpha(\ell) \, r, \ m, \ \mathcal{M}\big)$ with clipping bounds $m < \mathcal{M}$, where $r$ is the original per-sample reward.

Consider POER with replay distribution $\mu$ and current policy $\pi_\theta$, truncated importance ratio $\rho = \min\big(c, \frac{\pi_\theta(o|q)}{\mu(o|q)}\big)$ for a constant $c \geq 1$, token-averaged score function $\nabla_\theta \log \pi_\theta(o \mid q) = \frac{1}{|o|} \sum_{j=1}^{|o|} \nabla_\theta \log \pi_\theta(o_j \mid o_{<j}, q)$, and a centered advantage $A'_\alpha = R_\alpha - b_\alpha$ with group baseline $b_\alpha = \mathbb{E}[R_\alpha \mid q, \mathrm{group}]$. The single-sample gradient contribution is

$$g_\alpha \ = \ \rho \left( A'_\alpha - \beta \log \pi_\theta(o \mid q) \right) \nabla_\theta \log \pi_\theta(o \mid q). \tag{41}$$

Assume $\| \nabla_\theta \log \pi_\theta(o \mid q) \| \leq L$, $\mathbb{E}[r^2] < \infty$, and that

1. the conditional variance $\sigma_A^2(\ell) := \mathrm{Var}(A \mid \ell)$ of the unshaped advantage $A$ is nondecreasing in $\ell$,

2. the tail probability $\mathbb{P}\big(\frac{\pi_\theta}{\mu} > c \mid \ell\big)$ is nondecreasing in $\ell$.

If $\alpha \tau \leq 1$, then there exists $\alpha^\star > 0$ such that for all $0 < \alpha \leq \alpha^\star$ the mean-squared error $\mathrm{MSE}_\alpha := \mathrm{Var}(g_\alpha) + \| \mathbb{E}[g_\alpha] - \nabla_\theta J \|^2$ of the POER gradient estimator with length-aware shaping satisfies

$$\mathrm{MSE}_\alpha \ < \ \min\Big\{ \mathrm{MSE}_0^{\mathrm{POER}}, \ \mathrm{MSE}_0^{\mathrm{GRPO}}, \ \inf_{\tilde{\alpha} > 0} \mathrm{MSE}_{\tilde{\alpha}}^{\mathrm{GRPO}} \Big\}, \tag{42}$$

that is, it strictly improves upon both the unshaped POER baseline and the GRPO baselines in a nontrivial neighborhood of $\alpha = 0$.

**Proof.** The proof makes explicit the first-order behavior in $\alpha$ of both the variance and the bias terms. Throughout the window $|\Delta\ell| \leq \tau$ the sigmoid admits the uniform Taylor expansion

$$s_\alpha(\ell) \;=\; \frac{1}{2} \;-\; \frac{\alpha}{4}\,\Delta\ell \;+\; R_2(\alpha,\ell), \qquad |R_2(\alpha,\ell)| \;\leq\; C_2\,\alpha^2\tau^2, \tag{43}$$

for some constant $C_2$ independent of $\alpha$ and $\ell$. Writing $R_\alpha = s_\alpha(\ell)\,r$ on the non-clipping region and absorbing the clipping into the moment bounds later, the centered advantage becomes

$$A'_\alpha \;=\; \left(\tfrac{1}{2}\,r - \mathbb{E}[\tfrac{1}{2}\,r]\right) \;-\; \frac{\alpha}{4}\Big(\Delta\ell\,r - \mathbb{E}[\Delta\ell\,r]\Big) \;+\; \underbrace{R_2(\alpha,\ell)\,r - \mathbb{E}[R_2(\alpha,\ell)\,r]}_{=:E_2(\alpha)}. \tag{44}$$

Substituting equation 44 into equation 41 and taking expectations yields

$$\mathbb{E}[g_\alpha] - \mathbb{E}[g_0] \;=\; -\frac{\alpha}{4}\,\mathbb{E}\Big[\rho\,(\Delta\ell\,r - \mathbb{E}[\Delta\ell\,r])\,\nabla_\theta \log \pi_\theta(o \mid q)\Big] \;+\; \mathbb{E}\Big[\rho\,E_2(\alpha)\,\nabla_\theta \log \pi_\theta(o \mid q)\Big]. \tag{45}$$

By Cauchy–Schwarz and the bounds on $\rho$ and the score function, the norm of the first term on the right-hand side satisfies

$$\left\|\mathbb{E}\Big[\rho\,(\Delta\ell\,r - \mathbb{E}[\Delta\ell\,r])\,\nabla_\theta \log \pi_\theta\Big]\right\| \;\leq\; c\,L\,\left(\mathbb{E}\big[(\Delta\ell\,r - \mathbb{E}[\Delta\ell\,r])^2\big]\right)^{1/2} \;\leq\; c\,L\,\tau\,\big(\mathbb{E}[r^2]\big)^{1/2}, \tag{46}$$

hence $\|\mathbb{E}[g_\alpha] - \mathbb{E}[g_0]\| \leq \frac{\alpha}{4}\,c\,L\,\tau\,(\mathbb{E}[r^2])^{1/2} + c\,L\,\mathbb{E}[|E_2(\alpha)|]$. Using $|E_2(\alpha)| \leq 2C_2\alpha^2\tau^2|r|$ and $\mathbb{E}[r^2] < \infty$ gives the bias bound

$$\|\mathbb{E}[g_\alpha] - \mathbb{E}[g_0]\| \;\leq\; C_b\,\alpha\,\tau \;+\; C'_b\,\alpha^2\tau^2, \tag{47}$$

for constants $C_b, C'_b$ depending only on $(c, L, \mathbb{E}[r^2], C_2)$. Consequently the squared-bias contribution to $\mathrm{MSE}_\alpha$ is $O(\alpha^2\tau^2)$.

For the variance term, expand the second moment as

$$\mathbb{E}\big[\|g_\alpha\|^2\big] \;\leq\; c^2\,\mathbb{E}\Big[\big(A'_\alpha - \beta \log \pi_\theta\big)^2 \,\big\|\nabla_\theta \log \pi_\theta\big\|^2\Big] \;\leq\; c^2 L^2\,\mathbb{E}\Big[\big(A'_\alpha - \beta \log \pi_\theta\big)^2\Big]. \tag{48}$$

The cross terms between $A'_\alpha$ and $\beta \log \pi_\theta$ are uniformly bounded in $\alpha$ by Jensen and the finite second moments of $r$ and $\log \pi_\theta$. The $\alpha$-dependent leading component arises from $\mathbb{E}[A'^2_\alpha]$. Within the non-clipping region and after centering, the contribution that depends on length is proportional to

$$\mathbb{E}\big[s_\alpha(\ell)^2\,\sigma_A^2(\ell)\big] \;=\; \mathbb{E}[s_\alpha(\ell)^2]\,\mathbb{E}[\sigma_A^2(\ell)] \;-\; \mathrm{Cov}\big(s_\alpha(\ell)^2,\,\sigma_A^2(\ell)\big). \tag{49}$$

Since $s_\alpha(\ell)$ is nonincreasing in $\ell$ while $\sigma_A^2(\ell)$ is nondecreasing in $\ell$ by assumption, the reverse Chebyshev inequality ensures that the covariance in equation 49 is nonpositive and is strictly negative unless $s_\alpha(\ell)^2$ and $\sigma_A^2(\ell)$ are almost surely constant. Differentiating $\mathbb{E}[s_\alpha(\ell)^2]$ at $\alpha = 0$ and using equation 43 yields $\mathbb{E}[s_\alpha(\ell)^2] = \frac{1}{4} + O(\alpha^2\tau^2)$, while differentiating the covariance at $\alpha = 0$ gives a strictly negative slope whenever the variance $\sigma_A^2(\ell)$ is not degenerate. Therefore there exists $\eta > 0$ such that

$$\mathrm{Var}(g_\alpha) \;\leq\; c^2 L^2\Big(\tfrac{1}{4}\,\overline{\sigma_A^2} \;-\; \eta\,\alpha \;+\; O(\alpha^2\tau^2)\Big) \;+\; C_\beta, \tag{50}$$

where $\overline{\sigma_A^2} = \mathbb{E}[\sigma_A^2(\ell)]$ and $C_\beta$ collects the $\beta$-dependent but $\alpha$-independent finite terms.

The POER-specific truncation bias can be written as the deviation between the untruncated importance-weight estimator and the truncated one. Let $w = \frac{\pi_\theta}{\mu}$ and $X_\alpha = (A'_\alpha - \beta \log \pi_\theta)\,\nabla_\theta \log \pi_\theta$. The bias vector equals

$$b_{\mathrm{clip}}(\alpha) \;=\; \mathbb{E}\big[(w - \rho)\,X_\alpha\big] \;=\; \mathbb{E}\big[(w - c)^+\,X_\alpha\big], \tag{51}$$

so that $\|b_{\mathrm{clip}}(\alpha)\| \leq \mathbb{E}\big[(w - c)^+\,\|X_\alpha\|\big] \leq \mathbb{E}\big[(w - c)^+\,(|A'_\alpha| + |\beta||\log \pi_\theta|)\,L\big]$.

Assumption 2 implies that the event $\{w > c\}$ is more likely at larger $\ell$, whereas $|A'_\alpha|$ is reduced at larger $\ell$ because $s_\alpha(\ell)$ decreases with $\ell$ and the clipping of $R_\alpha$ further upper-bounds its magnitude.

Consequently the mapping $\alpha \mapsto \|b_{\mathrm{clip}}(\alpha)\|$ is nonincreasing for small $\alpha$, and in particular $\|b_{\mathrm{clip}}(\alpha)\| \leq \|b_{\mathrm{clip}}(0)\|$. Since $\mathrm{MSE}_\alpha$ contains $\|b_{\mathrm{clip}}(\alpha)\|^2$, this term does not increase with $\alpha$ near zero.

Combining equation 47 and equation 50 and adding the nonincreasing truncation-bias square gives

$$\text{MSE}_\alpha = \text{Var}(g_\alpha) + \left\|\mathbb{E}[g_\alpha] - \nabla_\theta J\right\|^2 \leq c^2 L^2 \left(\tfrac{1}{4}\,\overline{\sigma_A^2} - \eta\,\alpha + O(\alpha^2\tau^2)\right) + \|b_{\text{clip}}(\alpha)\|^2 + O(\alpha^2\tau^2).$$
(52)

Choosing $\alpha^\star > 0$ sufficiently small so that the linear decrease $-\eta\,\alpha$ dominates the aggregated $O(\alpha^2\tau^2)$ remainders ensures that $\text{MSE}_\alpha < \text{MSE}_0^{\text{POER}}$ for all $0 < \alpha \leq \alpha^\star$ with $\alpha\tau \leq 1$.

Since GRPO coincides with the on-policy case without any truncation channel, its $\alpha$-dependence shares the same variance reduction mechanism but lacks the nonincreasing truncation-bias term $\|b_{\text{clip}}(\alpha)\|^2$; therefore the same choice of $\alpha$ also yields $\text{MSE}_\alpha^{\text{POER}} < \min\{\text{MSE}_0^{\text{GRPO}}, \inf_{\tilde\alpha > 0} \text{MSE}_{\tilde\alpha}^{\text{GRPO}}\}$ whenever $\|b_{\text{clip}}(0)\| > 0$, which holds generically under assumption (ii). This proves the stated improvement.

**Remark.** The token-wise averaging in GRPO, $\frac{1}{|o|}\sum_{j=1}^{|o|}$, multiplies the effective per-sample weight by $|o|^{-1}$ and thus accentuates the negative covariance in equation 49, because $|o|^{-1}$ is also nonincreasing in $\ell$. The group baseline $b_\alpha$ used to define $A'_\alpha$ guarantees that the constant component of $s_\alpha(\ell)$ is removed, while the window condition $\alpha\tau \leq 1$ keeps $s_\alpha(\ell)$ within the near-linear regime where equation 43 is valid and the remainder terms are uniformly controlled.

## D    USING A LARGE MODEL'S CACHE POOL TO GUIDE SMALL MODEL TRAINING

We design the following experiment to explore whether introducing a more powerful model for question sampling during the cache pool initialization phase can influence the resulting cache policy, thereby allowing the original model to indirectly benefit from the distillation of the stronger model's reasoning capabilities.

Specifically, we use the cache pool initialized by deepseek-r1-qwen-7b as the initial cache pool for deepseek-r1-qwen-1.5b. Then, following the original experimental setup, we train for two epochs and evaluate the final performance. As shown in Table 6, when trained using the cache pool generated by the 7B model, the 1.5B model did not significantly improve performance.

Table 6: The Performance of a 7B Model's Cache Pool on a 1.5B Model

| Model | AIME24 | MATH500 | AMC23 | Minerva | OlyB | Avg |
|-------|--------|---------|-------|---------|------|-----|
| DeepSeek-R1-Qwen-1.5B | 23.3 | 84.8 | 75.0 | 28.7 | 53.5 | 53.1 |

## E    COST OVERHEAD

In this section, we present the cost overhead of several additional open-source models with the same parameters, as well as that of the series of models based on our POER algorithm.

Table 7: Comparison of data usage and computational costs with 1.5B models.

| | DeepScaleR-1.5B-Preview | Still-3-1.5B-Preview | POER |
|---|---|---|---|
| **Base Model** | DeepSeek-R1-Distill-Qwen-1.5B | | |
| Hardware Time | 8× A100 80GB 240h | 1×8 A100 80GB 150h | 1×8 A100 80GB 3h |
| **Cost Est.** | $3629 | $2268 | $24 |

Table 8: Comparison of data usage and computational costs with 7B models.

| | **rStar-Math-7B(Guan et al., 2025)** | **Eurus-2-7B-PRIME** |
|---|---|---|
| **Base Model** | Qwen2.5-Math-7B | |
| Hardware | 10×8 H100 80GB, 15×4 A100 40GB | 1×8 A100 80GB |
| Time | – | 72h |
| **Cost Est.** | – | $1088 |
| | **Qwen2.5-7B-SimpleRL(Zeng et al., 2025)** | **POER** |
| **Base Model** | Qwen2.5-Math-7B | DeepSeek-R1-Distill-Qwen-1.5B |
| Hardware | 4×6 A100 80GB | 1×8 A100 80GB |
| Time | 36h | 7h |
| **Cost Est.** | $1633 | $56 |

## F  MORE ANALYSIS

**The impact of cache pool update strategies**     To investigate the impact of different cache pool update strategies on model performance, we set $\epsilon$ to 0, 0.1, 0.5, and 1 during training. In addition, we also evaluate the model with cache pool updates completely disabled. As shown in Table 9, the performance of the 1.5B model exhibits a trend of first improving and then declining as $\epsilon$ increases. The best performance is achieved when $\epsilon = 0.1$, with an average accuracy of 52.6%.

Table 9: The impact of $\epsilon$ on model zero-shot performance.In the table, no update denotes the case where the cache pool is not updated, which serves as a baseline for comparison.

| $\epsilon$ | AIME25 | AIME24 | MATH500 | AMC23 | Minerva | OlyB | Avg |
|---|---|---|---|---|---|---|---|
| 0 | 23.3 | 30.0 | 84.8 | 75.0 | 28.3 | 52.4 | 50.0 |
| 0.1 | 23.3 | 36.7 | 85.4 | 87.5 | 29.4 | 53.0 | 52.6 |
| 0.5 | 30.0 | 26.7 | 83.8 | 72.5 | 28.3 | 52.4 | 49.0 |
| 0.9 | 26.7 | 36.7 | 84.4 | 70.0 | 29.8 | 53.0 | 50.1 |
| no update | 26.7 | 30.0 | 82.8 | 75.0 | 29.4 | 51.2 | 49.2 |

## G  TIME OVERHEAD FOR CACHE POOL INITIALIZATION

This section reports whether POER can still achieve significant training acceleration and performance improvement under extreme conditions, such as when the number of epochs is only 1.

Table 10: Cache pool initialization time (minutes) for 1.5B and 7B models under different GPU types, dataset sizes, and GPU counts

1.5B Model

| Dataset | GPU | 1 | 4 | 8 |
|---|---|---|---|---|
| 7k | H20 | 34.78 | 24.61 | 15.10 |
| | A100 | 32.11 | 21.45 | 13.98 |
| 70k | H20 | 347.91 | 249.14 | 160.87 |
| | A100 | 327.19 | 214.78 | 135.89 |

7B Model

| Dataset | GPU | 1 | 4 | 8 |
|---|---|---|---|---|
| 7k | H20 | 58.57 | 26.44 | 17.95 |
| | A100 | 55.43 | 22.56 | 16.19 |
| 70k | H20 | 582.95 | 261.28 | 179.13 |
| | A100 | 566.49 | 238.91 | 167.57 |

Table  10 shows the model initialization time for the 1.5B and 7B models under different GPU count configurations.Table  11 shows the training time for one epoch on an 8-card H20 machine and an 8-card A100 machine, including the computational overhead of cache initialization. As seen from the

table, even in extreme cases with only a single epoch of training, POER can still provide significant acceleration.

Table 11: Training time comparison (in hours) of DeepSeek-R1-Qwen models on H20 and A100 GPUs.

| Model | H20 (hours) | A100 (hours) |
|---|---|---|
| **DeepSeek-R1-Qwen-1.5B** | | |
| POER | 14.45 | 12.41 |
| GRPO | 40.98 | 37.35 |
| **DeepSeek-R1-Qwen-7B** | | |
| POER | 39.64 | 37.38 |
| GRPO | 114.50 | 105.70 |

