# OpenReview forum: "Policy Optimization with Experience Replay: Guiding Reasoning Models to Complete the Reasoning Path"
_ICLR.cc/2026/Conference — ICLR 2026 Conference Withdrawn Submission_

### Official Review · Reviewer_UGT9 · 2025-10-23

**Soundness:** 3
**Presentation:** 2
**Contribution:** 3
**Rating:** 6
**Confidence:** 3

**Summary:**

The paper identifies the substantial computational cost and instability of RLFT for LLMs as a key bottleneck, primarily due to the need to generate complete reasoning paths during the sampling phase. To address this, the authors propose a method that leverages a cache of previously generated high-quality answers. Instead of generating a response from scratch, it retrieves a cached answer for a given question, truncates the final tokens, and tasks the model with generating only the suffix. This approach is claimed to greatly reduce training time and improve stability. The authors also introduce Length-Aware Reward Shaping mechanism to encourage more concise reasoning. The method is evaluated on different model sizes across several math reasoning datasets, demonstrating significant reductions in training time and stable, competitive performance against baselines.

**Strengths:**

1. The work tackles a problem of high practical importance. The computational expense of RLFT is a major barrier to its widespread adoption and research. The claimed training time reductions of up to 90% are dramatic and, if robust, would represent a significant practical contribution to the field.

2. The paper's core concept is intuitive and clearly articulated. The motivation for avoiding full-path generation is well-established, and the proposed solution is easy to understand. Moreover, the solid and detailed theoretical analysis in this paper can explain the motivation of experience replay well.

3. The empirical evaluation is extensive. The authors validate their method on multiple model sizes and a range of standard reasoning benchmarks. The experiments are well-designed, including ablation studies on key hyperparameters and transferability of cache between different sizes of models.

**Weaknesses:**

1. Potential negative effect on the baseline methods. In Table 1, in some benchmarks, the performance of model with POER drops a little, showing the potential negative effect of POER on the reasoning performances of LLMs.

2. The experiments are exclusively focused on mathematical reasoning datasets.  These tasks often have highly structured reasoning paths where the correctness of a prefix strongly predicts the correctness of the full solution. This is a best-case scenario for POER. The method's viability is far less certain on more open-ended or creative tasks where diverse, valid reasoning paths exist. The approach implicitly assumes a single "good" prefix per problem, which is a limiting assumption that may not hold in other domains.

3. The proof of gradient stability in Appendix B essentially formalizes that conditioning on more information reduces variance.  However, it fails to address the more critical issue of bias. By constraining the sampling distribution, POER introduces a potentially significant bias, and this trade-off is never discussed. The reduced variance might simply be a byproduct of the policy learning a less diverse set of behaviors, which is potentially suboptimal.

**Questions:**

1. Given the heavy reliance on cached prefixes, how can you be sure the model is not simply overfitting to these specific reasoning paths?

2. Could you provide a more thorough explanation for why providing a 1.5B model with superior reasoning paths from a 7B model yields no significant benefit?

3. The performance of POER seems highly dependent on the quality of the initial cache. How does the algorithm perform if the initial model fails to find correct solutions for a significant portion of the training set, leaving parts of the cache empty or filled with incorrect paths?

---

> ### Author Response · Authors · 2025-11-24
> **Response Q1**
>
> Dear Reviewer,
>
> Thank you very much for raising these three highly insightful questions.
>
> ### 1. Regarding concerns about overfitting to cached prefixes
>
> This is a crucial issue. We designed specific mechanisms and validation methods to ensure that the model is learning **how to reason**, rather than merely memorizing paths from the cache.
>
> **Dynamic Truncation Strategy**: We do not provide the model with fixed prefixes. During the sampling and generation phase, we introduce a dynamic truncation mechanism. For a historical answer $a_k$ retrieved from the cache, we truncate the last $m$ tokens, where $m$ is randomly sampled from a uniform distribution $[0, L]$ $m \sim \mathcal{U}[0,1,...,L]$. This means that, even for the same question, the “prompt prefix” seen by the model varies in length and content across training steps. This randomness forces the model to learn to continue reasoning from different intermediate points of the reasoning path, enhancing robustness rather than rote memorization.
>
> **Zero-Shot Generalization Validation**: The most direct evidence comes from our evaluation. When evaluating on standard test sets such as AIME and MATH500, we adopt a **zero-shot** setting, where no cache is used and the model must independently generate the full reasoning path from scratch. Experimental results show that POER performs better than or comparable to baseline methods (e.g., the 1.5B model achieves 51.7% on average, surpassing GRPO’s 47.8%). If the model were merely overfitting to cached prefixes, it would perform poorly in zero-shot testing without any prefix guidance—but this is clearly not the case.
>
> **Continuous Cache Updates**: The cache is not static. We continuously update it during training using an $\epsilon$-greedy strategy with newly generated high-reward answers. This ensures that the guided paths themselves evolve alongside the improving model capabilities.

---

> ### Author Response · Authors · 2025-11-24
> **Response Q2**
>
> ### 2. On why using a 7B model to guide the 1.5B model (Appendix D) did not yield significant gains
>
> In **Appendix D**, we specifically discuss this “large-to-small” experimental setup. In short, we used the cache initialized from the DeepSeek-R1-Qwen-7B model to train the 1.5B model.
>
> Intuitively, one might expect that reasoning paths from a stronger model would provide better “distillation” benefits. However, experimental results show that the 1.5B model achieves an average score of 53.1% when using the 7B cache, which is only a modest improvement over its best performance using its own cache (51.7%).
>
> Our analysis is as follows:
>
> * **Model capacity bottleneck**: The main limitation likely lies in the capacity of the 1.5B model itself. Even if the 7B model provides more complex or higher-level reasoning paths, the 1.5B model may be constrained by its parameter size and unable to fully understand or imitate these advanced reasoning patterns.
>
> * **Inability to perform initial exploration**: The core logic of POER relies on using prefixes of correct reasoning paths to reduce unproductive search space. However, the 1.5B model cannot learn initial exploration solely from 7B reasoning path prefixes; it can only learn to complete subsequent parts of the path.
>
> * **RFT supervision is weaker than SFT supervision**: During SFT, we can distill knowledge from the large model to the small model via the autoregressive loss. In contrast, during RFT, the loss function does not provide token-level supervision comparable to autoregressive logits, making it difficult for the small model to acquire new knowledge in this way.

---

> ### Author Response · Authors · 2025-11-24
> **Response Q3**
>
> ### 3. Regarding the robustness to initial cache quality and the “cold start” problem
>
> POER indeed relies on the existence of correct-answer paths in the cache. To address the concern you raised about low cache quality due to the initial model’s limited problem-solving ability, our algorithm incorporates the following mechanisms:
>
> * **Extensive sampling during initialization**: During the cache initialization phase, we are not limited to a single sample. Although, for efficiency, we only perform one round in our experiments, the vllm framework allows parallel multiple inferences for the same problem to capture as many correct solutions as possible within the current model’s capability (Equation 1 defines sampling from the initial policy:
> $
> a \sim \pi_{\theta_0}(\cdot | s)
> $.
>
>
> * **Error-correction capability via the $\epsilon$-Greedy update mechanism**: Even if the initial cache contains suboptimal paths or exploration is limited during subsequent training, we employ an $\epsilon$-greedy update strategy. This means that when updating the cache, there is a probability of $1-\epsilon$ to randomly select a suboptimal response, or if the model generates a new higher-reward path during exploration, it will replace the old one.
>
> * **Gradual recovery of exploration ability**: We analyze this in Section 5 (Figure 7). Experiments show that although POER initially has lower exploration ability than GRPO, as epochs progress and the cache is updated, the model gradually discovers more diverse and high-quality solution paths, steadily improving performance and eventually surpassing GRPO.

---

> ### Author Response · Authors · 2025-11-29
>
> Dear Reviewer UGT9,
>
> Thank you very much for your valuable comments on our paper.
>
> We would like to follow up on the response we previously posted, in which we addressed all of your comments in detail.
>
> If you have any remaining concerns, or if you think that any of the clarifications in our response would benefit from further explanation, please feel free to let us know — we would be glad to provide additional details.
>
> If you believe that the revised manuscript and rebuttal have sufficiently addressed your main concerns, we would greatly appreciate it if you could reflect this in your final evaluation.
>
> Sincerely,
> The Authors

---

### Official Review · Reviewer_DqSL · 2025-10-28

**Soundness:** 2
**Presentation:** 2
**Contribution:** 2
**Rating:** 2
**Confidence:** 2

**Summary:**

The paper aims to reduce training time while improving training stability. The paper proposes the assumption that during reinforcement fine-tuning, the model only needs to generate part of the reasoning process. Instead of always generating full reasoning paths, POER trains the model by generating suffixes of the reasoning path using experience caching.

**Strengths:**

1. The paper focuses on an interesting question.
2. The paper conducts ablation and additional experiments to understand the impact of length-aware reward design and the sensitivity of hyperparameters.
3. Experimental details are provided for reproduction

**Weaknesses:**

1. I think some related work and baselines are missing. For example, experience replay has been proposed by other papers recently [1-3].
2. The experiments only use Deepseek-r1-qwen-distill-1.5b and Deepseek-r1-qwen-distill-7b as base models. The models are already well fine-tuned for the reasoning task. It's unclear if the method will still work using less powerful base models.
3. The writing can be improved for better clarity. For instance, Table 1 is somewhat confusing. Demonstrating that GRPO/DAPO improves over the base model does not directly support the main claim. We care more about how GRPO/DAPO compares with GRPO/DAPO + POER. The same issue applies to the “w/o R” results. In addition, the main message of Figure 3 is unclear and should be explained more explicitly.
4. The results appear to be quite sensitive to hyperparameters. As mentioned in the paper:  "The evaluation results are shown in Table 5: under a fixed L, performance first improves as α increases and then declines." This indicates that the method’s stability and generality may depend strongly on fine-tuning specific parameters.
5. Also, the paper says "POER demonstrates lower exploration ability compared to GRPO, and the gap between the two methods
gradually widens as max_length increases." This seems to reveal a clear limitation of the proposed method.
[1] Zhang, Hongzhi, et al. "Rlep: Reinforcement learning with experience replay for llm reasoning." arXiv preprint arXiv:2507.07451 (2025).
[2] Zhang, Xuechen, et al. "BREAD: Branched Rollouts from Expert Anchors Bridge SFT & RL for Reasoning." arXiv preprint arXiv:2506.17211 (2025).
[3] Dou, Shihan, et al. "Improving rl exploration for llm reasoning through retrospective replay." arXiv preprint arXiv:2504.14363 (2025).

**Questions:**

See weakness

---

> ### Author Response · Authors · 2025-11-24
> **Response Q1**
>
> Dear Reviewer,
>
> Thank you very much for your detailed and highly constructive feedback.
>
> ### 1. On the missing related work
>
> We sincerely appreciate your pointing out these recent papers on Experience Replay. We acknowledge that our related work section omitted these important contemporary studies. The initial version of our work was completed in April 2025, at a time when the application of Experience Replay in RL4LLM was still very limited. In the current submission, we unfortunately overlooked updating this part of the description—this was our mistake.
>
> In the revised version of the paper, we will add a discussion of these works in the “Related Work” section (Section 2). We will clearly distinguish POER from these methods: the key novelty of POER lies in combining **guided suffix generation** with **length-aware rewards**, which not only leverage experience replay for stabilizing training but also explicitly optimize for inference efficiency.
>
> Regarding the benchmarks, we adopted the most widely used benchmarks in several contemporaneous studies [1–3]. While these benchmarks may not be completely comprehensive, in our view they sufficiently meet the requirements for evaluating model capabilities.
>
> [1] [https://arxiv.org/abs/2503.18892](https://arxiv.org/abs/2503.18892)
> [2] [https://arxiv.org/abs/2501.04519](https://arxiv.org/abs/2501.04519)
> [3] [https://arxiv.org/abs/2503.16219](https://arxiv.org/abs/2503.16219)

---

> ### Author Response · Authors · 2025-11-24
> **Response Q2**
>
> ### 2. Regarding the choice of base models
>
> You pointed out that we primarily evaluated our method on the relatively strong DeepSeek-R1-Distill model and raised concerns about whether the method would remain effective on weaker models. This is a very reasonable concern. To address this, we conducted additional experiments on other models. Due to time constraints, we report results on DeepSeek-R1-Distill-Llama-8B and the weaker Qwen2.5-7B-Math model.
>
> #### **Results on DeepSeek-R1-Distill-Llama-8B**
>
> | Model            | AIME24 | AIME2025 | MATH-500 | AMC23 | Minerva | olympaidbench |
> | ---------------- | ------ | -------- | -------- | ----- | ------- | ------------- |
> | Distill-Llama-8B | 0.4    | 0.3      | 0.81     | 0.775 | 0.425   | 0.537         |
> | +GRPO (w/o R)    | 0.433  | 0.367    | 0.852    | 0.8   | 0.431   | 0.561         |
> | +POER            | 0.433  | 0.4      | 0.830    | 0.825 | 0.431   | 0.554         |
> | +GRPO (w/ R)     | 0.4    | 0.367    | 0.830    | 0.8   | 0.447   | 0.541         |
> | +POER            | 0.466  | 0.433    | 0.866    | 0.825 | 0.439   | 0.569         |
> | +DAPO (w/o R)    | 0.433  | 0.4      | 0.826    | 0.8   | 0.431   | 0.571         |
> | +POER            | 0.433  | 0.4      | 0.878    | 0.825 | 0.451   | 0.571         |
> | +DAPO (w/ R)     | 0.433  | 0.367    | 0.864    | 0.449 | 0.775   | 0.546         |
> | +POER            | 0.467  | 0.4      | 0.888    | 0.85  | 0.461   | 0.574         |
>
> #### **Results on Qwen2.5-7B-Math**
>
> | Model           | AIME24 | AIME2025 | MATH-500 | AMC23 | Minerva | olympaidbench | AVG   |
> | :-------------- | :----- | :------- | :------- | :---- | :------ | :------------ | :---- |
> | Qwen2.5-7B-Math | 0.2    | 0.1      | 0.836    | 0.625 | 0.371   | 0.416         | 0.425 |
> | +GRPO (w/o R)   | 0.233  | 0.133    | 0.862    | 0.750 | 0.401   | 0.427         | 0.468 |
> | +POER           | 0.2    | 0.133    | 0.844    | 0.675 | 0.401   | 0.421         | 0.446 |
> | +GRPO (w/ R)    | 0.233  | 0.133    | 0.840    | 0.675 | 0.395   | 0.423         | 0.450 |
> | +POER           | 0.233  | 0.233    | 0.866    | 0.775 | 0.401   | 0.433         | 0.490 |
> | +DAPO (w/o R)   | 0.267  | 0.133    | 0.846    | 0.675 | 0.389   | 0.431         | 0.457 |
> | +POER           | 0.233  | 0.2      | 0.850    | 0.7   | 0.398   | 0.427         | 0.468 |
> | +DAPO (w/ R)    | 0.233  | 0.2      | 0.864    | 0.650 | 0.381   | 0.431         | 0.460 |
> | +POER           | 0.267  | 0.2      | 0.854    | 0.7   | 0.417   | 0.431         | 0.478 |

---

> ### Author Response · Authors · 2025-11-24
> **Response Q3**
>
> ### 3. Regarding the clarity of figures
>
> We fully agree with your observation. The presentation of Table 1 and Figure 3 can indeed be improved to more intuitively support our key arguments, and we will make the necessary revisions in the next version.

---

> ### Author Response · Authors · 2025-11-24
> **Response Q4**
>
> ### 4. Regarding hyperparameter sensitivity ($\alpha$)
>
> You pointed out the sensitivity of the results to the length penalty coefficient $\alpha$. As shown in Table 5 and the related analysis, the performance indeed exhibits a trend of first increasing and then decreasing as α increases. This occurs because $\alpha$ controls the strength of the penalty on reasoning length: a too-small α fails to encourage the model to generate concise reasoning paths, while a too-large α may lead the model to sacrifice accuracy in order to shorten the length. Nevertheless, we consider this a reasonable trade-off. Our results also show that even under different settings (e.g., $\alpha$ = 0.01 or 0.1), POER generally outperforms the unweighted baseline.

---

> ### Author Response · Authors · 2025-11-24
> **Response Q5**
>
> ### 5. Regarding the limitation of weaker exploration
>
> You noted a limitation of POER in terms of exploration: “POER demonstrates lower exploration ability compared to GRPO.” We acknowledge that this is indeed a notable limitation, but it is actually a deliberate trade-off we made for **efficiency** and **stability**. While GRPO performs full-path sampling with strong exploration, it suffers from extremely high variance and is prone to collapse (as shown in Figure 4, GRPO experiences length collapse around 200 steps).
>
> In contrast, although POER exhibits weaker initial exploration, we mitigate this through a **cache update strategy**. As shown in Figure 7, with increasing epochs (and continuous cache updates), POER’s exploration capability and overall performance steadily improve, eventually catching up with or even surpassing GRPO. This indicates that POER is not incapable of exploration, but rather explores in a controlled and gradual manner.

---

> ### Author Response · Authors · 2025-11-29
>
> Dear Reviewer DqSL,
>
> Thank you very much for your valuable comments on our paper.
>
> We would like to follow up on the response we previously posted, in which we addressed all of your comments in detail.
>
> If you have any remaining concerns, or if you think that any of the clarifications in our response would benefit from further explanation, please feel free to let us know — we would be glad to provide additional details.
>
> If you believe that the revised manuscript and rebuttal have sufficiently addressed your main concerns, we would greatly appreciate it if you could reflect this in your final evaluation.
>
> Sincerely,
> The Authors

---

### Official Review · Reviewer_ibC2 · 2025-10-28

**Soundness:** 1
**Presentation:** 2
**Contribution:** 1
**Rating:** 2
**Confidence:** 4

**Summary:**

Existing reinforcement learning (RL) fine-tuning methods typically require generating a complete reasoning trajectory from the beginning, which introduces substantial computational overhead. To address this challenge, the paper proposes POER (Policy Optimization with Experience Replay) — a reinforcement fine-tuning algorithm that leverages partial reasoning prefixes from a cache of previous trajectories to generate sufficient completions. In addition, the method introduces a length-aware reward to encourage more effective policy gradient updates. The paper provides both theoretical analysis, showing that POER yields a more stable training process, and empirical results demonstrating that across six reasoning benchmarks, POER significantly reduces training time while achieving comparable or slightly improved accuracy relative to GRPO and DAPO.

**Strengths:**

1. The training speed improvements are quantitatively convincing.

2. The paper is well-organized, providing a clear step-by-step methodological exposition. Figures and tables effectively illustrate both the conceptual workflow and empirical results.

**Weaknesses:**

1. The idea of learning from partial rollouts has been extensively discussed in prior literature [1], and the use of length-based rewards [2] is also not a new technique. However, the paper does not adequately discuss the differences or connections between these existing works and the proposed method.

2. The average sequence lengths used for both training and evaluation appear to be quite short, which limits comparability across methods. For instance, DeepScaleR-1.5B-Preview [3], trained with pure GRPO on DeepSeek-R1-Distill-Qwen-1.5B, achieves 43.1% on AIME2024, 87.8 on MATH, 73.6 on AMC2023, and 30.2 on MinervaMath, higher than the results reported here. This discrepancy likely stems from shorter evaluation lengths, but the paper does not clarify this. As a result, POER might make the model more myopic, performing optimistically well on short-length reasoning but losing extrapolation capability for longer chains.

3. Moreover, the pass@k performance of POER is substantially worse than the baseline, suggesting that the method sacrifices exploration even with limited training. This raises concerns about its long-term training dynamics and ultimate performance ceiling.

4. The assumption: “for the same question, a reasoning path that reaches the correct answer more concisely should be rewarded with a higher value” doesn’t seem to be supported by any evidence. From a reinforcement learning perspective, excessively concise reasoning paths may encourage memorization rather than adaptive exploration. In contrast, longer reasoning chains often include more trial-and-error behavior, potentially fostering more generalizable reasoning patterns that perform better on harder or unseen problems.

[1] Qu, Yuxiao, et al. "Optimizing test-time compute via meta reinforcement fine-tuning." arXiv preprint arXiv:2503.07572 (2025).

[2] Yeo, Edward, et al. "Demystifying long chain-of-thought reasoning in llms." arXiv preprint arXiv:2502.03373 (2025).

[3] Luo, Michael, et al. DeepScaleR: Surpassing O1-Preview with a 1.5B Model by Scaling RL. 2025, pretty-radio-b75.notion.site/DeepScaleR-Surpassing-O1-Preview-with-a-1-5B-Model-by-Scaling-RL-19681902c1468005bed8ca303013a4e2.

**Questions:**

1. The paper claims that POER yields a more stable training process than GRPO by comparing variance metrics. However, the theoretical proof of variance reduction (particularly the decomposition shown in Equation (11)) hinges on assumptions that do not hold in the presented setting. Specifically, in POER, $a_k$ is off-policy, sampled from a cache composed of prior policies. Therefore, the conditional expectation $\mathbb{E}_{\left(q_k, a_k\right) \mid q_k}$ in the proof is not taken under the same distribution as the current gradient estimator. This discrepancy invalidates the steps from Eq. (11) → (13) → (18) unless the cache distribution is explicitly modeled and corrected, e.g., via importance sampling.

2. Without such corrections, the POER gradient estimator becomes biased, making a “variance reduction” claim theoretically unfounded. Empirically, the results, reported only up to four epochs, are insufficient to convincingly demonstrate improved stability over GRPO or DAPO, especially for longer training runs where such bias may accumulate.

---

> ### Author Response · Authors · 2025-11-24
> **Response Q1**
>
> Dear Reviewer,
>
> You have raised very insightful theoretical concerns. We greatly appreciate your rigorous examination of our mathematical derivations and experimental design.
>
> ### 1. Regarding the distribution mismatch and bias in theoretical derivations
>
> We fully acknowledge your point: in a strict mathematical sense, since the samples in the cache $\mathcal{C}$ come from historical policies (i.e., off-policy), the variance decomposition in Equation (11) indeed makes a simplifying assumption by treating the cache policy $\pi_{\mathcal{C}}$ as an approximation of the current policy $\pi_{\theta}$.
>
> However, we would like to defend the design of POER from the following two perspectives:
>
> * **As a "beneficial bias" regularization mechanism:**
>   Although off-policy sampling does introduce a statistical “bias,” this is actually a core design intention of POER. As stated in the paper, the goal of POER is to restrict the exploration space $\pi_{\theta}$ to a smaller, high-quality subspace that has been validated.
>   The variance reduction claimed in Equations (8) and (35) essentially arises from narrowing the integration domain of the gradient estimator from the full space (high-entropy, containing many erroneous trajectories) to a local neighborhood around successful trajectories (low-entropy). Even though this makes the gradient estimator biased with respect to the original objective in theory, in practice this bias serves as a strong regularizer, preventing the model from losing direction in early-stage exploration when rewards are sparse.
>
> * **$\epsilon$-Greedy updates mitigate distribution shift:**
>   To reduce the discrepancy between historical and current policies, we do not use a static cache. Instead, the cache is dynamically updated after each epoch using an $\epsilon$-greedy strategy. This means that most of the paths in the cache are generated by relatively recent policies (when $\epsilon$ is small, we tend to retain high-scoring answers produced by the current policy). This mechanism ensures that $\pi_{\mathcal{C}}$ does not lag far behind $\pi_{\theta}$, making the approximation in Appendix B acceptable for local update steps.
>
> Regarding the decomposition from (11) to (13):
>
> For Equation (12), we rewrite it based on $E_{q_k}$:
>
> $$
> E_{q_k}[Var(\cdot|q_k)] \geq E_{q_k}[E_{(q_k,a_k)|q_k}[Var(\cdot|(q_k,a_k))]]
> $$
>
> which leads to:
> $$
> E_{q_k}[Var(\cdot|q_k)] \geq E_{(q_k,a_k)}[Var(\cdot|(q_k,a_k))].
> $$
>
> At this point, for the element-wise statistics in (18), we can obtain its weakened condition:
> $$
> E_{q_k} \big|\big|\nabla\theta \pi_\theta(a_g \mid q_k)\big|\big|2
> \ge
> E_{q_k,a_k} \big|\big|\nabla\theta \pi_\theta(a_g \mid q_k, a_k)\big|\big|_2
> $$
>
> Intuitively, conditioning the gradient estimator on the concrete cached prefix $(q_k,a_k)$ removes the additional randomness arising from resampling different answers for the same question $q_k$. Under the replay distribution $\mu(q,a)$, the law of total variance implies that the average conditional variance of the policy value (and, via the first-order approximation $(\mathrm{Var}(\pi_\theta) \approx |\nabla_\theta \pi_\theta|_2^2)$) is no larger when conditioning on $(q_k,a_k)$ than when conditioning only on $q_k$. Therefore, gradient estimates based on $(q_k,a_k)$ fluctuate less across samples, leading to a more stable optimization process in POER compared to GRPO.

---

> ### Author Response · Authors · 2025-11-24
> **Response Q2**
>
> ### 2. Persuasiveness of the 4-Epoch Experimental Results
>
> In the Reinforcement Learning Fine-Tuning, model collapse typically occurs very quickly. As we demonstrated in **Section 4.4 Stability Analysis**, the baseline method GRPO, under the same settings, exhibited a severe drop in response length (from ~1600 tokens to <800 tokens) and performance deterioration after only about **200 steps**—far fewer than 4 epochs. In contrast, POER maintained stable response lengths and performance throughout the full 4 epochs (over 1,000 steps) of training.
>
> The concern about cumulative error, if it exists, should theoretically manifest as gradual performance degradation over time. However, Figure 7 shows that as epochs increase (with the cache continuously updated), POER’s Pass@N performance steadily improves. This indicates that the bias we introduced effectively guides the model toward better local optima rather than causing divergence. In fact, for high-epoch settings, RFT algorithms without experience replay—i.e., without this introduced bias—tend to show performance decay, as noted in studies such as DAPO.
>
> To provide stronger evidence, we report results from 12-epoch experiments during the rebuttal stage.  Due to time constraints, we trained each method only once and report the results from that single run.
>
> | Model                     | AIME25 | AIME24 | MATH500 | AMC23 | Minerva | OlyB  | Avg   |
> | ------------------------- | ------ | ------ | ------- | ----- | ------- | ----- | ----- |
> | **DeepSeek-R1-Qwen-1.5B** | 0.167  | 0.288  | 0.822   | 0.629 | 0.265   | 0.433 | 0.434 |
> | + GRPO(w/o R)             | 0.133  | 0.233  | 0.688   | 0.650 | 0.235   | 0.411 | 0.392 |
> | + **POER**                | 0.267  | 0.300  | 0.832   | 0.750 | 0.287   | 0.511 | 0.491 |
> | + GRPO(w/ R)              | 0.133  | 0.233  | 0.656   | 0.600 | 0.213   | 0.442 | 0.380 |
> | + **POER**                | 0.267  | 0.367  | 0.852   | 0.800 | 0.298   | 0.512 | 0.516 |
> | + DAPO(w/o R)             | 0.200  | 0.200  | 0.674   | 0.600 | 0.263   | 0.462 | 0.400 |
> | + **POER**                | 0.233  | 0.233  | 0.866   | 0.825 | 0.291   | 0.522 | 0.495 |
> | + DAPO(w/ R)              | 0.200  | 0.200  | 0.652   | 0.575 | 0.214   | 0.472 | 0.386 |
> | + **POER**                | 0.233  | 0.300  | 0.870   | 0.850 | 0.301   | 0.522 | 0.513 |
>
> | Model                   | AIME25 | AIME24 | MATH500 | AMC23 | Minerva | OlyB  | Avg   |
> | ----------------------- | ------ | ------ | ------- | ----- | ------- | ----- | ----- |
> | **DeepSeek-R1-Qwen-7B** | 0.433  | 0.555  | 0.928   | 0.900 | 0.445   | 0.674 | 0.656 |
> | + GRPO(w/o R)           | 0.375  | 0.475  | 0.880   | 0.825 | 0.402   | 0.612 | 0.596 |
> | + **POER**              | 0.425  | 0.475  | 0.924   | 0.900 | 0.423   | 0.677 | 0.636 |
> | + GRPO(w/ R)            | 0.325  | 0.425  | 0.872   | 0.800 | 0.388   | 0.594 | 0.565 |
> | + **POER**              | 0.500  | 0.600  | 0.942   | 0.900 | 0.437   | 0.673 | 0.678 |
> | + DAPO(w/o R)           | 0.375  | 0.475  | 0.884   | 0.825 | 0.406   | 0.621 | 0.596 |
> | + **POER**              | 0.475  | 0.525  | 0.942   | 0.900 | 0.427   | 0.649 | 0.653 |
> | + DAPO(w/ R)            | 0.350  | 0.500  | 0.864   | 0.825 | 0.392   | 0.583 | 0.587 |
> | + **POER**              | 0.475  | 0.625  | 0.948   | 0.950 | 0.431   | 0.645 | 0.665 |

---

> ### Author Response · Authors · 2025-11-29
>
> Dear Reviewer ibC2,
>
> Thank you very much for your valuable comments on our paper.
>
> We would like to follow up on the response we previously posted, in which we addressed all of your comments in detail.
>
> If you have any remaining concerns, or if you think that any of the clarifications in our response would benefit from further explanation, please feel free to let us know — we would be glad to provide additional details.
>
> If you believe that the revised manuscript and rebuttal have sufficiently addressed your main concerns, we would greatly appreciate it if you could reflect this in your final evaluation.
>
> Sincerely,
> The Authors

---

### Official Review · Reviewer_QhcC · 2025-10-30

**Soundness:** 3
**Presentation:** 2
**Contribution:** 2
**Rating:** 4
**Confidence:** 4

**Summary:**

This paper proposes Policy Optimization with Experience Replay (POER), a plug-and-play reinforcement learning fine-tuning algorithm. The main contribution of this work is to use partially correct reasoning process hints to guide the LLM RL training to improve the training efficiency and reach a more stable training.

**Strengths:**

1. The motivation and the logic of the algorithm are reasonable.
2. The presentation of the method is clear.
3. The experiments evaluate 6 datasets and 2 different model sizes, which makes this part more comprehensive.

**Weaknesses:**

1. The model family is limited, only containing DeepSeek models.
2. The task is limited, only containing mathematical reasoning tasks.
3. There is no study about the generalization ability of the models after this training method.
4. The usage of LLM writing is obvious; although the authors clarify the usage, it is better to rewrite some parts, such as those parts using '-', and also some obvious typos, such as model names.
5. While the method is generally reasonable, it is hard to encounter one problem several times during large-scale training in practice, which I personally think will limit the practicality of this method.

**Questions:**

1. Is it possible to evaluate some other reasoning tasks, such as coding and commonsense reasoning?
2. Is it possible to include some study about other model families, such as Qwen3 and Llama?
3. Is it possible to add some studies about out-of-domain tasks?
4. Is there a possibility that this method introduces a distribution shift during training because of the usage of cache and the special reward shaping to diversify the reward of sampled generations?

---

> ### Author Response · Authors · 2025-11-24
> **Response Q1**
>
> Dear Reviewer,
>
> Thank you for raising these excellent questions regarding experimental breadth, model generality, and theoretical depth. We will address your suggestions one by one and explain our considerations in the context of the current paper.
>
> ### 1. Regarding evaluation on coding and commonsense reasoning tasks
>
> This is a very valuable suggestion, which could further validate the generality of POER.
>
> In this study, we primarily evaluate our models on six standard mathematical reasoning datasets (AIME25, AIME24, MATH500, AMC23, Minerva, OlympiadBench). These datasets were chosen because the majority of related work also evaluates on these benchmarks. Aligning with prior work allows us to provide comparable results.
>
> Although the current experiments focus on the mathematics domain, POER is fundamentally a plug-and-play, policy-agnostic algorithm. Its core mechanism—leveraging experience replay to complete reasoning paths—is not limited to any specific domain. In principle, POER can be applied to coding tasks or commonsense reasoning as long as the tasks require multi-step reasoning and provide verifiable rewards.
>
> However, since the training data used in our experiments are mathematics reasoning datasets, our evaluation is naturally limited to these mathematical benchmarks.

---

> ### Author Response · Authors · 2025-11-24
> **Response Q2**
>
> ### 2. Regarding extending to other model families such as Qwen3 and Llama
>
> In the paper, we selected DeepSeek-R1-Qwen-Distill-1.5B and 7B as our base models. These models were chosen because they were representative among the available open-source reasoning models at the time and had already undergone a cold-start phase of reinforcement learning, making them suitable for direct RL fine-tuning. Although Qwen3 has advanced reasoning capabilities, it is an MoE model, and its RFT stage code differs from that of dense models. Due to time constraints, we were unable to implement this part in the short term.
>
> For the purpose of this rebuttal, we report results using DeepSeek-R1-Llama-Distill and Qwen2.5-7B-Math. In the revised version of the paper, we plan to include results for Qwen3 as well.
>
> #### **Results on DeepSeek-R1-Llama-Distill**
>
> | Model            | AIME24 | AIME2025 | MATH-500 | AMC23 | Minerva | olympaidbench |
> | ---------------- | ------ | -------- | -------- | ----- | ------- | ------------- |
> | Distill-Llama-8B | 0.4    | 0.3      | 0.81     | 0.775 | 0.425   | 0.537         |
> | +GRPO (w/o R)    | 0.433  | 0.367    | 0.852    | 0.8   | 0.431   | 0.561         |
> | +POER            | 0.433  | 0.4      | 0.830    | 0.825 | 0.431   | 0.554         |
> | +GRPO (w/ R)     | 0.4    | 0.367    | 0.830    | 0.8   | 0.447   | 0.541         |
> | +POER            | 0.466  | 0.433    | 0.866    | 0.825 | 0.439   | 0.569         |
> | +DAPO (w/o R)    | 0.433  | 0.4      | 0.826    | 0.8   | 0.431   | 0.571         |
> | +POER            | 0.433  | 0.4      | 0.878    | 0.825 | 0.451   | 0.571         |
> | +DAPO (w/ R)     | 0.433  | 0.367    | 0.864    | 0.449 | 0.775   | 0.546         |
> | +POER            | 0.467  | 0.4      | 0.888    | 0.85  | 0.461   | 0.574         |
>
> #### **Results on Qwen2.5-7B-Math**
>
> | Model           | AIME24 | AIME2025 | MATH-500 | AMC23 | Minerva | olympaidbench | AVG   |
> | :-------------- | :----- | :------- | :------- | :---- | :------ | :------------ | :---- |
> | Qwen2.5-7B-Math | 0.2    | 0.1      | 0.836    | 0.625 | 0.371   | 0.416         | 0.425 |
> | +GRPO (w/o R)   | 0.233  | 0.133    | 0.862    | 0.750 | 0.401   | 0.427         | 0.468 |
> | +POER           | 0.2    | 0.133    | 0.844    | 0.675 | 0.401   | 0.421         | 0.446 |
> | +GRPO (w/ R)    | 0.233  | 0.133    | 0.840    | 0.675 | 0.395   | 0.423         | 0.450 |
> | +POER           | 0.233  | 0.233    | 0.866    | 0.775 | 0.401   | 0.433         | 0.490 |
> | +DAPO (w/o R)   | 0.267  | 0.133    | 0.846    | 0.675 | 0.389   | 0.431         | 0.457 |
> | +POER           | 0.233  | 0.2      | 0.850    | 0.7   | 0.398   | 0.427         | 0.468 |
> | +DAPO (w/ R)    | 0.233  | 0.2      | 0.864    | 0.650 | 0.381   | 0.431         | 0.460 |
> | +POER           | 0.267  | 0.2      | 0.854    | 0.7   | 0.417   | 0.431         | 0.478 |

---

> ### Author Response · Authors · 2025-11-24
> **Response Q3**
>
> ### 3. Regarding out-of-distribution (OOD) tasks
>
> * **Current generalization tests**: We trained on the Open-RS dataset and performed zero-shot evaluations on six benchmarks, including AIME, MATH500, Minerva, and OlympiadBench. While these benchmarks are still primarily in the mathematical domain, datasets such as **OlympiadBench**  and **Minerva** already represent some distribution shifts relative to the training data, thereby testing the model’s generalization ability.
>
> * **Zero-shot performance**: Our experimental results show that POER generally achieves zero-shot performance on these benchmarks that is comparable to or better than GRPO. This indicates that the model does not overfit to the specific training distribution, but instead has learned a general reasoning capability through path completion.

---

> ### Author Response · Authors · 2025-11-24
> **Response Q4**
>
> ### 4. Regarding whether the cache and specialized rewards introduce distribution shift
>
> This is a very profound theoretical question. Our perspective is that POER does introduce a certain form of shift, but this shift acts as a **beneficial constraint** rather than a harmful distribution shift.
>
> * **Regularizing the gradient space**: As described in the paper, POER incorporates historical sampled response trajectories as constraints during subsequent sampling. This constraint effectively limits the exploration space of the policy $\pi_{\theta}$ during training, thereby regularizing the gradient descent space.
>
> * **Reducing variance and improving stability**: This intervention on the distribution does not harm training; instead, it reduces the variance of gradient estimates. As we demonstrate in Equation (8) and Appendix B,
> $
> Var(||∇_θ J_P||_2) \le Var(||∇_θ J_G||_2)
> $. This means that this “shift” actually stabilizes the training process and prevents the typical collapses seen in GRPO caused by excessive randomness (e.g., response length collapse).
>
> * **Mitigating locality issues**: Traditional GRPO suffers from severe locality problems because it only compares samples within the current batch. POER, by introducing an experience cache and leveraging the external cache policy $\pi_{\mathcal{C}}$, approximates the policy at the time of update, providing a “global context.” This mechanism biases the distribution toward historically verified high-quality paths, which is precisely the intended outcome.
>
> * **Complementarity of length-aware rewards**: Regarding the reward design, we show that length-aware reward shaping is complementary to POER. Due to shared prefixes reducing intra-batch sample diversity, introducing length-based reward differences helps maintain meaningful gradient signals. This targeted reward design is meant to accommodate the distribution characteristics induced by the cache mechanism, thereby achieving improved performance gains.

---

> ### Author Response · Authors · 2025-11-29
>
> Dear Reviewer QhcC,
>
> Thank you very much for your valuable comments on our paper.
>
> We would like to follow up on the response we previously posted, in which we addressed all of your comments in detail.
>
> If you have any remaining concerns, or if you think that any of the clarifications in our response would benefit from further explanation, please feel free to let us know — we would be glad to provide additional details.
>
> If you believe that the revised manuscript and rebuttal have sufficiently addressed your main concerns, we would greatly appreciate it if you could reflect this in your final evaluation.
>
> Sincerely,
> The Authors

---

### Note · Authors · 2026-01-05

I have read and agree with the venue's withdrawal policy on behalf of myself and my co-authors.